

# Recalibration of low-cost air pollution sensors: Is it worth it?

Paul Gäbel[1] and Elke Hertig[1]

[1]Regional Climate Change and Health, Faculty of Medicine, University of Augsburg, Universitätsstraße 2, 86159 Augsburg, Germany

*Correspondence to*: Paul Gäbel (paul.gaebel@med.uni-augsburg.de)

**Abstract.** The appropriate period of collocation of a low-cost air sensor (LCS) with reference measurements is often unknown. Previous low-cost air sensor studies have shown that due to sensor ageing and seasonality of environmental interferences periodical sensor calibration needs to be performed to guarantee sufficient data quality. While the limitations are well-established it is still unclear how often a recalibration of a sensor needs to be carried out. In this study, we aim to demonstrate

how frequently widely used air sensors for the relevant air pollutants $O_3$ and $PM_{2.5}$ by two manufacturers (Alphasense and Sensirion) should be recalibrated. Sensor calibration functions were built using Multiple Linear Regression, Ridge Regression, Random Forest and Extreme Gradient Boosting. We use state-of-the-art test protocols for air sensors provided by the United States Environmental Protection Agency (EPA) and the European Committee for Standardization (CEN) for evaluative guidance. We conducted a yearlong collocation campaign at an urban background air and climate monitoring station next to

the University Hospital Augsburg, Germany. LCS were exposed to a wide range of environmental conditions, with air temperatures between -10 and 36 °C, relative air humidity between 19 and 96 % and air pressure between 937 and 983 hPa. The ambient concentration ranges for $O_3$ and $PM_{2.5}$ were up to 83 ppb and 153 µg m$^{-3}$, respectively. For the baseline single training of 5 months, the calibrated $O_3$ and $PM_{2.5}$ sensors were able to reflect the hourly reference data well during the training ($R^2$: $O_3$ = 0.92–1.00; $PM_{2.5}$ = 0.93–0.98) and the following test period ($R^2$: $O_3$ = 0.93–0.97; $PM_{2.5}$ = 0.84–0.93). Additionally,

the sensor errors were generally acceptable during the training (RMSE: $O_3$ = 0.80–4.35 ppb; $PM_{2.5}$ = 1.45–2.51 µg m$^{-3}$) and the following test period (RMSE: $O_3$ = 3.62–5.84 ppb; $PM_{2.5}$ = 2.04–3.02 µg m$^{-3}$). By investigating different recalibration cycles using a pairwise calibration strategy, our results indicate that a regular in-season recalibration is required to obtain the highest quantitative validity for the analysed low-cost air sensors, with monthly recalibrations appearing to be the most suitable approach. In contrast, an extension of the training period for the calibration models had only a minor overall impact on

improving the low-cost air sensors' ability to capture temporal variations in observed $O_3$ concentrations and $PM_{2.5}$ concentrations. The measurement uncertainty of the calibrated $O_3$ LCS and $PM_{2.5}$ LCS were able to meet the data quality objective (DQO) for indicative measurements for different calibration models. Compared to one-time pre-deployment sensor calibration, in-season recalibration can broaden the scope of application for a LCS (indicative measurements, objective estimation, non-regulatory supplemental and informational monitoring).



## 1 Introduction

Low-cost sensors (LCS) form an interesting approach for monitoring air pollution in a denser network than currently available due to the cost of regular fixed measurement stations. Basically, they are smaller, consume less power, are cheaper and therefore more accessible than regular monitoring devices for air pollution (Jiao et al., 2016; Lewis et al., 2018; Li et al., 2020; Peltier et al., 2021; Schäfer et al., 2021). This underlines why there is an interest among researchers, governments, businesses

and individuals in using LCS for air quality monitoring in different settings, e.g. citizen science, mobile and stationary monitoring (in for instance urban or remote locations), urban planning, personal exposure science or education (Williams et al., 2019; Mahajan and Kumar, 2020; Mahajan et al., 2020; Peltier et al., 2021; Okure et al., 2022; Hassani et al., 2023; Malings et al., 2024). This interest has led researchers to develop their own custom-built air quality monitoring systems equipped with LCS (Mueller et al., 2017; Cross et al., 2017; Gäbel et al., 2022), which can be more widely used in the aforementioned settings

due to lower costs.

Nevertheless, those sensors also have their disadvantages. At present they do not fulfill the stringent requirements for regulatory measurements provided by high-quality air pollutant monitoring systems for instance used by governments to monitor the exceedance of health relevant thresholds for air pollutants like ozone ($O_3$), nitrogen oxides ($NO_x$), particulate

matter ($PM_{2.5}$, $PM_{10}$), carbon monoxide (CO), and sulfur dioxide ($SO_2$) (Castell et al., 2017; Wesseling et al., 2019; Schäfer et al., 2021). Major issues with LCS are their short operating life, the lack of long-term stability due to sensor ageing, interferences, cross-sensitivities, the need of calibration functions to adjust their bias as well as transforming LCS output into meaningful units (Lewis et al., 2018; Peltier et al., 2021; Concas et al., 2021; Carotenuto et al., 2023). Hence reference measurements are needed. The inter-sensor unit variability of LCS is another issue, where a calibration function derived

through training data for a LCS is usually not by default transferable. LCS data of a unit can be quite unique, when compared to data of another unit of the same model (Moltchanov et al., 2015; Gäbel et al., 2022; Bittner et al., 2022). However, good-performing sensors can act as non-regulatory supplemental and informational monitoring (NSIM) devices, where data quality control procedures must be explored to investigate the individual LCS and its data to classify it as a NSIM device (Duvall et al., 2021a; Duvall et al., 2021b; Malings et al., 2024). It boils down to the question, if the selected air sensor is a good fit for

its planned purpose (Diez et al., 2022). Snyder et al. (2013) summarized the essence of the problem in one sentence: "Data of poor or unknown quality is less useful than no data since it can lead to wrong decisions".

Uniform evaluation and comparison methods for LCS are incentivized by a growing market, which offers a greater supply of more refined low-cost air sensors. The lack of standardized procedures was pointed out in the literature in the recent years (Rai

et al., 2017; Karagulian et al., 2019; Williams et al., 2019; Duvall et al., 2021a), therefore there is an initiative of multiple organizations to develop test programs and test protocols like the Environmental Protection Agency (EPA) of the United States or European Committee for Standardization (CEN) (Duvall et al., 2021a; Duvall et al., 2021b; CEN/TS 17660-1:2021; CEN/TS



17660-2:2024). The development of test programs by organizations, which are also recognized by governmental bodies, is an important achievement. They create a foundational framework to collect comparable harmonized metrics to assess LCS data

quality and therefore helping to develop a standardized quality assessment to ultimately justify the use of LCS in defined areas of interest in air pollution monitoring. Hence it is a further step for establishing reliable low-cost air quality networks within the regulatory monitoring system for air quality worldwide. Using target metrics from these test programs to better understand the performance of air sensors is not yet commonly practiced in studies evaluating LCS in different settings.

One important aspect are recalibrations of LCS after their initial (on-site) calibration using reference monitors, which is an important point in network management to guarantee longer-term data quality (Concas et al., 2021; Carotenuto et al., 2023). However, most of the recent studies doing longer-term field campaigns using LCS networks for air quality monitoring show in their methods no recalibration strategy to mitigate the effect of sensor ageing and thus to enhance the LCS measurement output under a quantitative point of view (Jayaratne et al., 2020; Petäjä et al., 2021; Mohd Nadzir et al., 2021; Bílek et al.,

2021; Raheja et al., 2022; Kim et al., 2022; Collier-Oxandale et al., 2022; Okure et al., 2022; Connolly et al., 2022). For instance, the official warranted operating lifespan of the commonly used electrochemical LCS NO2-B43F by the company Alphasense is only 2 years (Alphasense, 2024a) or even lower according to Li et al. (2021). They investigated the long-term degradation of electrochemical Alphasense $NO_2$ sensors in the field and found evidence, that those sensors could already malfunction after 200 days. Furthermore Kim et al. (2022) calibrated Alphasense $NO_2$ sensors based on a 6-month collocation

using regulatory monitoring devices at a rural traffic site. 1.5 years later Kim et al. (2022) did a second collocation experiment with the same sensors at the same site using their original calibration functions from the first collocation. They found a significant deterioration in sensor performance during the second collocation. It was also discussed that due to time-varying effects of environmental interferences (e.g. air temperature, relative humidity), sensor performance can vary with season (Ratingen et al., 2021; Peters et al., 2022). For these reasons, LCS recalibration intervals of less than 1 year and methods for

regular LCS data quality checks using regulatory monitoring devices need to be explored whether those sensor devices are supposed to be used in lengthy measurement campaigns to assess air quality. At present it is quite unknown how regular LCS need to be recalibrated and the number of publications investigating varying calibration periods isn't exhaustive due to the lack of long-term collocation experiments in the available literature. Generally, studies which investigate varying recalibration periods look only at a specific air pollutant sensor targeting one air pollutant and don't consider state-of-the-art test programs

for LCS to categorize their results in frameworks provided by organizations, which are officially recognized by governmental authorities.

We investigated different recalibration cycles for commonly used LCS for $NO_2$, $O_3$, CO and $PM_{2.5}$ using metrics and target values provided by EPA and CEN (Duvall et al., 2021a; Duvall et al., 2021b; CEN/TS 17660-1:2021; CEN/TS 17660-2:2024).

We conducted the investigation during a one-year on-site collocation experiment in the city of Augsburg, Germany. An in-depth investigation was done for LCS measuring $O_3$ and $PM_{2.5}$. Due to test site limitations affecting the ability to classify LCS

for NO₂ and CO according to CEN, these air substances could only be classified using the EPA test protocol for gas sensors. Two Atmospheric Exposure Low-Cost Monitoring (AELCM) boxes developed by the Chair for Regional Climate Change and Health of the University of Augsburg where mounted next to the Atmospheric Exposure Monitoring Station (AEMS) for air

substances and meteorological variables operated by the same chair. The boxes included the LCS for the mentioned air pollutants while the latter provided the reference measurements in the present study.

This work is organized as follows. The section about materials and methods describes the infrastructure used for the collocation experiment (AELCM, AEMS) and the methodology behind our sensor calibration strategy and its evaluation. The "Results

and Discussion" section focuses on the environmental conditions and pollution concentrations observed during the collocation experiment, the performance of the introduced LCS calibration models under different recalibration cycles and the potential implications of our findings for LCS networks. The concluding remarks can be found in the last section.

## 2 Materials and methods

### 2.1 AELCM Sensor box

Two advanced Atmospheric Exposure Low-Cost Monitoring (AELCM) sensor boxes were used, denoted as AELCM009 and AELCM010. The custom-built devices were developed by the Chair for Regional Climate Change and Health at the University of Augsburg. A detailed description and performance check of the first version of the low-cost measurement unit can be found in our previous study (Gäbel et al., 2022). The upgraded AELCM units measured air quality and meteorological parameters, namely O₃ (Alphasense Ox-B431), NO₂ (Alphasense NO2-B43F), CO (Alphasense CO-B4), PM₂.₅ (Sensirion AG SPS30) as

well as humidity and air temperature (Bosch BME280) (Bosch Sensortec, 2015; Sensirion, 2023; Alphasense, 2024a, c, b). In this study the air pollution sensors were denoted as AS-B431, AS-B43F, AS-B4 and SAG-SPS30. The upgrade of the AELCM boxes with respect to the previous study were related to the switch to electrochemical gas sensors from Alphasense, which exclusively measured the earlier mentioned gaseous air substances. The upgrades also involved the increase of the measurement frequency for each AELCM sensor from 10 seconds to every 4 seconds. A code rework on the Arduino

microcontroller board made it possible to measure on a higher temporal resolution.

There were two reasons for the switch to the Alphasense sensors: Gäbel et al. (2022) concluded, that the digital gas sensor models DGS-NO2 and DGS-CO from SPEC Sensors showed no capability to capture the given concentrations at a measurement station according to the coefficient of determination and the Spearman rank correlation. The SPEC DGS-O3

units performed well overall but showed a high inter-sensor unit variability.





## 2.2 Collocation with AEMS

The AELCM units were mounted on a fence right next to the AEMS. The AEMS is a high-quality air and climate measurement station located next to the University Hospital Augsburg in Germany (48°23.04' N, 10°50.53' E). The station can be classified as an urban background station. Federal roads are in the south and east, respectively 850 meters and 1200 meters located away from the station. A highway road is 3600 meters located away in the North. Industrial areas relative to the station location are located further away, in the south-east and the north-east of Augsburg. Regular station measurements of varying concentrations of CO, $NO_2$ and $PM_{2.5}$ due to local traffic and local industry depend highly on circulation patterns favouring an air flow from those sources towards the city as well as on the day and daytime, where factors like commuting play an important role.

The regulatory grade air measurement instruments are from the company HORIBA. Reference measurements of $O_3$, $NO_2$, CO and $PM_{2.5}$ were conducted using the instruments APOA-370, APNA-370, APMA-370 and APDA-372, in that order. The HORIBA instruments for gaseous air pollutants are also used by the Bavarian Environment Agency for official air pollution monitoring in Bavaria (Bayerisches Landesamt für Umwelt, 2019). The weather station WS600-UMB mounted to the station provided measurements for meteorological variables. Further details about the AEMS can be found in the study of Gäbel et al. (2022).

The collocation took place from January 2022 until January 2023. The model training period for the LCS was between 11[th] of January 2022 till 10[th] of June 2022. The testing period for the LCS recalibration experiment started at the 10[th] of June 2022. The experiment ended between the 6[th] and 11[th] of January 2023 depending on the LCS. The end is individual for each LCS model unit, because of individual missing values in the reference measurements for each air pollutant caused by regular station maintenance work or due to power grid tests at the University Hospital. The aim of the collocation was the assessment of the benefit of regular recalibrations against single calibration. The latter used solely the above-mentioned training period for model training. Performance metrics and their recommended target values given by test programs and test protocols by EPA and CEN were considered to assess the influence of a recalibration procedure on LCS performance (Duvall et al., 2021a; Duvall et al., 2021b; CEN/TS 17660-1:2021; CEN/TS 17660-2:2024).

## 2.3 Data treatment

The collocation experiment involving AELCM009 and AELCM010 started originally on the 10[th] of January 2022. The equipped low-cost air sensors have a stabilization phase after powering them. Only after their stabilization phase the LCS output is eligible for measurements of their respective target pollutant (Gäbel et al., 2022). The first 24 hours of all LCS data were thus removed and not considered for this study. The AS-B431 is an LCS, which measures $O_3$ and $NO_2$ (Alphasense, 2024c). For the correct measurement of ambient $O_3$ using an AS-B431 unit, data of a LCS measuring $NO_2$ is required for the O3 calibration model. For this purpose, we used an AS-B43F unit. The modelled calibration functions for the estimation of $O_3$



included by default the LCS output of both Alphasense sensor units. The Alphasense sensors provided voltages as measurement outputs by default. Like Bigi et al. (2018), we calculated the net voltage of every Alphasense sensor based on the difference

between the working and auxiliary electrodes. The calculated net voltages became an input for the modelled calibration functions next to the meteorological variables air temperature and relative humidity, which affect the LCS output as environmental interferences.

The system time of the AELCM units (UTC) were adjusted to the system time of the AEMS (CET). LCS measurements and

reference measurements given by the AEMS were aggregated to hourly means. $PM_{2.5}$ measurements were also aggregated to daily means, which was required for the performance evaluation of the low-cost particulate matter sensor SAG-SPS30 based on the technical specification developed by CEN (CEN/TS 17660-2:2024, 2024) and the test protocol developed by EPA (Duvall et al., 2021a). The missing values in the air pollution reference data were caused by regular maintenance and device malfunctions. The missing values in the meteorological data were due to device malfunctions of the weather station. The LCS

measurement data for each AELCM unit was nearly complete, with very few missing values, similar to the data in Gäbel et al. (2022). 100 % and at least 80 % of the data had to be available for the hourly aggregation of reference measurements of gaseous air pollutants and meteorological variables, respectively. For the calculation of the key performance metric in the technical specification by CEN (CEN/TS 17660-2:2024, 2024), the minimum data capture of the SAG-SPS30 was set to 90 %. Therefore, the daily means of $PM_{2.5}$ based on reference and LCS data were only valid if at least 90 % of the hourly averages

were available within a 24 h period. Note, that the data completeness criterion is less strict in the particulate matter sensor test protocol by EPA. There, the daily mean $PM_{2.5}$ concentration is calculated on at least 75 % of hourly averages within a 24 h period (Duvall et al., 2021a).

For gaseous air constituents the devices in the AEMS and the model calibrated LCS devices provided measurements in the

unit parts per billion (ppb). Hence for the calculation of mass concentrations the hourly aggregated meteorological measurements of the integrated weather station of the AEMS were used. Mass concentrations were needed for the performance evaluation of ambient air quality sensors for gaseous pollutants following the technical specification developed by CEN (CEN/TS 17660-1:2021, 2021). The SAG-SPS30 for the measurement of particulate matter provides outputs in mass concentrations by default. We didn't use the low-cost meteorological data of the Bosch BME280 to calculate the mass

concentrations for the LCS gas measurement devices, because the measurements are highly biased due to solar radiation. The bias stems from solar heating of the AELCM units, which could not be mitigated by the integrated fan, as it causes an exchange of air between the inside and outside, failing to reduce the heating effect. It is planned to upgrade the AELCM units with radiation shields in the future to reduce the effect of solar radiation on the low-cost meteorological measurements.





## 2.4 LCS calibration and model tuning

We built and evaluated four regression models (calibration models) for each LCS to estimate air pollution levels based on their data output (hourly means) considering environmental influences on sensor output and reference measurements (AEMS). The regression models were based on Multiple Linear Regression (MLR), Ridge Regression (RR), Random Forest (RF) and Extreme Gradient Boosting (XGB). Moving forward we will call these calibration models. Every calibration model consisted of a target variable to be predicted, and features used for prediction. As target we defined the ambient air pollutant concentration

of a specific air substance ($AEMS_{O3}$, $AEMS_{NO2}$, $AEMS_{CO}$, $AEMS_{PM2.5}$). As features used for prediction, we used the raw LCS output. The LCS output can be classified into the net voltages given by each Alphasense sensor ($V_{OX10}$, $V_{OX09}$, $V_{NO210}$, $V_{NO209}$, $V_{CO10}$, $V_{CO09}$), the mass concentrations given by each particulate matter sensor SPS30 ($SPS30_{10}$, $SPS30_{09}$) and the air temperatures ($T_{10}$, $T_{09}$) and relative humidities ($RH_{10}$, $RH_{09}$) provided by each BME280.

We chose MLR models because MLR is still the most common basic approach in the literature to develop calibration models for LCS (Karagulian et al., 2019). In this paper, we used MLR with the setup as in Gäbel et al. (2022) extended by an interaction term according to Bigi et al. (2018) as the reference calibration approach next to machine learning approaches, i.e. Random Forest (Breiman, 2001) and Extreme Gradient Boosting (Chen and Guestrin, 2016). Also Ridge Regression was applied (Friedman et al., 2010), which includes an approach to adjust for collinearity between model features. For the development of

the MLR models, we considered the usual MLR statistical model assumptions and checks, including the inspection of the residuals as well as the findings from the work of Bigi et al. (2018) and Hasan et al. (2023). Based on the findings of Bigi et al. (2018), we've used net voltages and a term for the interaction between net voltages and air temperature as features. Furthermore, Hasan et al. (2023) found a calibration model performance improvement using $O_3$ and $NO_2$ sensors, when they added the output of a low-cost CO sensor as a feature. We considered both findings for our own calibration models. The

selected features and targets for every calibration model can be found in Table 1.

**Table 1.** Model variables for the development of the calibration functions based on Multiple Linear Regression (MLR), Ridge Regression (RR), Random Forest (RF) and Extreme Gradient Boosting (XGB).

| Calibration Model | O₃ Model (Features / Target) | NO₂ Model (Features / Target) | CO Model (Features / Target) | PM₂.₅ Model (Features / Target) |
|---|---|---|---|---|
| MLR | $V_{OX}$, $V_{NO2}$, $V_{CO}$, RH, T, $V_{OX} * T$ / $AEMS_{O3}$ | $V_{NO2}$, $V_{CO}$, RH, T, $V_{NO2}*T$ / $AEMS_{NO2}$ | $V_{CO}$, RH, T, $V_{CO} * T$, $\frac{(V_{CO})^2 - 1}{2}$ / $AEMS_{CO}$ | SPS30, RH, T, log(SPS30) / $\log(AEMS_{PM2.5})$ |
| RR | $V_{OX}$, $V_{NO2}$, $V_{CO}$, RH, T / $AEMS_{O3}$ | $V_{NO2}$, $V_{CO}$, RH, T / $AEMS_{NO2}$ | $V_{CO}$, RH, T / $AEMS_{CO}$ | SPS30, RH, T / $AEMS_{PM2.5}$ |
| RF | $V_{OX}$, $V_{NO2}$, $V_{CO}$, RH, T / $AEMS_{O3}$ | $V_{NO2}$, $V_{CO}$, RH, T / $AEMS_{NO2}$ | $V_{CO}$, RH, T / $AEMS_{CO}$ | SPS30, RH, T / $AEMS_{PM2.5}$ |
| XGB | $V_{OX}$, $V_{NO2}$, $V_{CO}$, RH, T / $AEMS_{O3}$ | $V_{NO2}$, $V_{CO}$, RH, T / $AEMS_{NO2}$ | $V_{CO}$, RH, T / $AEMS_{CO}$ | SPS30, RH, T / $AEMS_{PM2.5}$ |






The development of the calibration models for the LCS data of both AELCM units using Random Forest, Extreme Gradient Boosting and Ridge Regression had the following steps: (1) Pre-processing of data given by the AEMS (Reference) and AELCM units (LCS) according to Sect. 2.3 and merging the data by hour; (2) Tuning of selected model hyperparameters during the first 5 months of the collocation period using the repeated holdout method (10 evaluation periods), random search

as search strategy and the root-mean-squared error (RMSE) as performance metric; (3) Applying the best hyperparameter configuration to the calibration model, and training it using a single calibration period (first 5 months of the collocation period) or an extended calibration period (further training). For step (2) and step (3) the package mlr3 in the statistics software R was used (Lang et al., 2019). The selected and tuned model hyperparameters for Random Forest, Extreme Gradient Boosting and Ridge Regression can be found in the supplement (Table S3).


The search strategy random search describes a random value selection in a pre-defined interval for each to be tuned model hyperparameter in an independent manner (Bergstra and Bengio, 2012; Becker et al., 2024). We selected random search as the search strategy for its simplicity and the possibility to use mixed search spaces (using numeric and integer hyperparameters) (Becker et al., 2024). Becker et al. (2024) also mention that random search is often the better choice to produce more unique

values per hyperparameter compared to grid search under the circumstance that certain hyperparameters only offer a minimal impact on model performance compared to others. Therefore, random search offered us a safer option to realize a meaningful hyperparameter tuning in a reasonable timeframe considering multiple models and LCS.

An out-of-sample method (OOS, Gäbel et al. (2022)) using a repeated holdout strategy for the tuning process was chosen to
obtain robust estimates. Summarizing this method, a random point t in time (e.g., 30 April 2022 12:00:00 CET) of the time series ts was chosen to separate the training and evaluation data. The previous window with reference to t comprising 60 % of ts was used for training and the following window of 10 % of ts was used for testing. For 10 repetitions, we received 10 randomly chosen dates t, which separated the training and evaluation sets. The sizes of the training and evaluation sets depended on the length of the available LCS time series and reference data. As mentioned in step (2), for the hyperparameter

tuning process we used the first 5 months of data per LCS during the collocation period. Finally, considering the average RMSE based on 10 evaluation periods, we chose the final hyperparameter configuration for each LCS calibration model. The hyperparameter tuning process was unique for each LCS calibration model. No generalized model for a specific sensor unit was developed.

## 2.5 Key aspects for exploring a pairwise calibration strategy

Low-cost air sensors are measurement instruments which need treatment. This treatment involves the consideration of regular maintenance work in a post-deployment setting involving recalibration and data quality assurance (Peltier et al., 2021; Concas et al., 2021). However, calibration of LCS requires immense effort and is resource-intensive in general. Carotenuto et al. (2023)



concluded that the comparison of LCS measurements against those from official reference stations for in situ calibration is often recommended in the scientific literature. Continuous and independent access to high quality equipment (e.g. laboratory,

monitoring station) for reference measurements would be ideal to establish and maintain low-cost air measurement networks but it is rather hard to achieve. Therefore, maintainers of LCS networks are either forced to rely on their established pre-deployment calibration functions (single calibration) or to find alternative, advanced network calibration methods to calibrate sensors in situ on a regular basis. Both usually rely on the measurement infrastructure of a third party in some form (e.g. local environmental agency). Alternative network calibration methods are for instance blind calibration, opportunistic and

collaborative calibration and calibration transfer (Maag et al., 2018; Concas et al., 2021), which increase the level of methodical complexity compared to a more traditional pairwise calibration strategy (Delaine et al., 2019). Latter is usually deemed unfeasible as a network calibration strategy in other literature as demonstrated in the following paragraph. A collocation calibration represents a pairwise calibration method.

It is argued by Mueller et al. (2017), that a collocation calibration using a reference measurement station is time consuming and that the infrastructure for that approach must be available in the first place. Broday et al. (2017) highlighted the impracticality of relying on collocations for regular LCS calibration and that in situ calibration methods could make the widespread use of LCS air pollution networks possibly more likely. Furthermore, regular recalibration using a collocation calibration hinders a continuous data collection in situ, because in situ measurements are interrupted to calibrate LCS (Broday

et al., 2017; Kizel et al., 2018). In this study we explore these issues through a calibration methodology, which involves a pairwise calibration strategy. Moreover, we analysed if less but more regularly calibrated LCS and less complex calibration methods (e.g. collocation) using a continuous stream of high-quality reference measurements can be an option to establish easier to manage (but smaller) LCS networks for long-term in situ measurements.

In most air sensor studies aiming at establishing a long-term low-cost air quality monitoring network, a pairwise calibration strategy is not seen as an option due to the focus on establishing spatially dense LCS networks. The resources for pairwise calibration are often not available and the method is considered resource-intensive, thus current and possibly future studies will not explore this method in an in-depth manner. This tendency is seen in the main recommendations delivered by other scientific papers (Carotenuto et al., 2023). Indeed, a continuous data collection in situ is an obstacle when a collocation

calibration is applied. This can be avoided by using a pair of LCS devices in situ. We explored using two AELCM units with the same sensor configuration for one location. One AELCM unit, which requires recalibration can be replaced with its partner AELCM unit. It must be noted that, while continuous, it creates a somewhat inhomogeneous measurement time series because the same location is alternately measured with two AELCM units.



## 2.6 Single training vs. extended training

A single training period (ST) represented a continuous time frame for model calibration. In this work an extended training period (ET) referred to a non-continuous time frame for model calibration, which was longer than the former. Non-continuous meant, that there were gaps of defined length between blocks of continuous data. Together these blocks formed the training data used for training the final calibration model. We also investigated the influence of the length of gaps on the model performance. As a baseline for reference, we used the model trained on the single, shorter training period. This approach helped us to examine the overall benefit of longer training periods on model performance considering that sensors degrade over time. Also, we wanted to study if shorter gaps influence the model performance considering the seasonal variability of air pollution and that sensor performance can vary with season due to time-varying effects of environmental interferences.

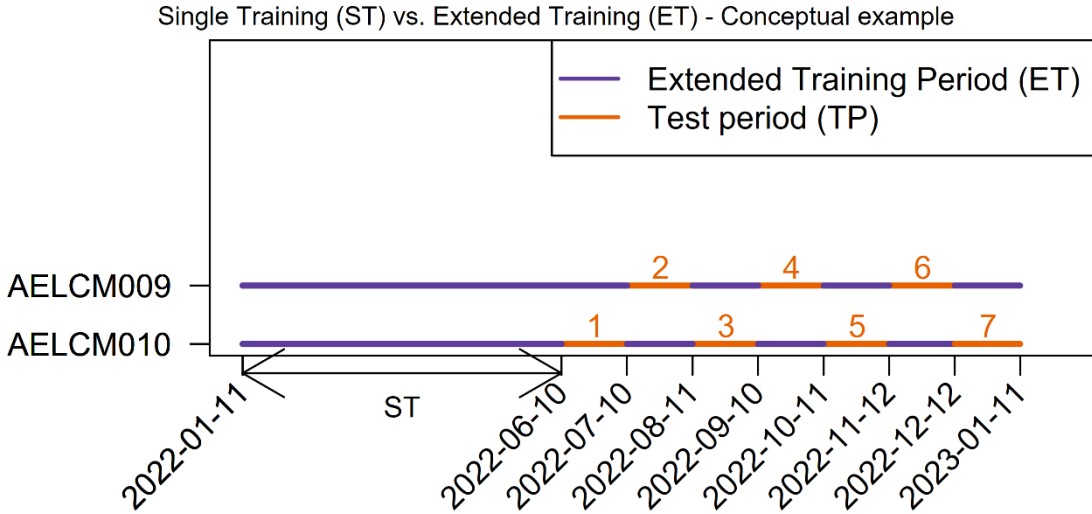

**Figure 1.** Schematic representation of the pairwise calibration strategy using two LCS measurement systems (AELCM009 and AELCM010) showing the single training period (ST, 11 January–10 June 2022) and the extended training period (ET) as well as the numbered one-month test periods (TP) for each LCS measurement system.

The outline of the approach is shown in Fig. 1. Since the primary goal of an air quality monitoring system equipped with LCS is to collect continuous measurements from a location outside a station site used for collocation calibration, we simulated the use of two calibrated LCS measurement systems alternately in the field. These two LCS measurement systems were represented through AELCM009 and AELCM010. Using both AELCM units, we received a continuous time series of in situ air pollution measurements. These in situ measurements are represented through the test periods (TP) in Fig. 1. By merging the blocks of continuous data (TP1 to TP7), we created a continuous time series in the field. Individual calibration models for each AELCM unit were trained using a ST period or an ET period. ST used approximately 5 months of hourly reference data and LCS data to train a final calibration model for each individual LCS. ET offered more training data across different seasons,



which reflects the aspect of regular LCS recalibration using reference monitors at a collocation site to guarantee long-term data quality. The ST served as a reference to investigate whether there is an actual benefit in extending the training period.

Figure 1 shows ET lengths of 1 month and the testing data blocks. We experimented with a length of 1, 2 and 3 months to study the influence on the model performance. In a LCS network setting, an ET length of 3 months would mean, that an AELCM unit would take in situ measurements for 3 months before being replaced by another calibrated AELCM unit. Therefore, the former unit can be relocated to the collocation site for 3 months to extend its training data and to quality check its data before switching places again with the latter unit. Please note, that we were restricted by the overall collocation

campaign length of 1 year. Selecting 5 months for the ST period, as shown in Fig. 1, resulted in seven months being available to fit the following data blocks, which were defined by the ET lengths. Two- and three-month ETs created a remainder of 1 training month at the end of the measurement campaign, which we considered as well for the ET to not waste training data.

For the ET setup all training data blocks were considered for training a calibration model. Thus, we performed an a posteriori
evaluation of the introduced pairwise calibration strategy including the introduced calibration models and ET lengths based on different performance metrics.

## 2.7 Performance metrics and target values

To benchmark the effect of using an ET approach compared to a ST approach, we mostly looked at commonly used and recommended performance metrics in LCS studies (Karagulian et al., 2019; Concas et al., 2021) and target values provided by
EPA and CEN (Duvall et al., 2021a; Duvall et al., 2021b; CEN/TS 17660-1:2021; CEN/TS 17660-2:2024). These performance metrics are the root-mean-square error (RMSE), mean absolute error (MAE), relative expanded uncertainty (REU), coefficient of determination ($R^2$), spearman rank correlation (Rs) as well as the regression slope and intercept. Here, a simple linear regression between model calibrated LCS data and AEMS reference data provide the slope and intercept (Duvall et al., 2021a; Duvall et al., 2021b). Most of the mentioned metrics are commonly used to describe LCS calibration model performance in
regards of bias, noise, linearity and error (Karagulian et al., 2019; Duvall et al., 2021a; Duvall et al., 2021b; Yatkin et al., 2022; Diez et al., 2022).

We aimed to analyse the consequences of ET by using a cohesive view of performance metrics and target values, introduced through state-of-the-art test programs. A major challenge for potential end-users of LCS is to interpret the calculated
performance metrics and thus to infer if a LCS is a good fit for an intended application (Diez et al., 2022). Recognized organizations linked with governmental bodies like CEN and EPA started to develop frameworks in the form of test protocols, which can be used to check the suitability of LCS for air quality monitoring applications. We used the performance metrics and associated categorizations given by state-of-the-art test programs as a reference to contextualize our study results.



However, we emphasize that due to methodological differences, our testing framework for air sensors does not fully align with those of EPA and CEN.

So far, the EPA offers target values for $O_3$, $NO_2$, CO, $SO_2$, $PM_{2.5}$ and $PM_{10}$ air sensors through their testing protocols. According to the EPA, the introduced performance metrics and their corresponding target values are the result of the current state of knowledge, based on, for example, literature reviews, findings from other organizations that conduct routine sensor evaluations and EPAs own expertise in sensor evaluation research (Duvall et al., 2021a; Duvall et al., 2021b). The current EPA test protocols include target values for the RMSE, $R^2$, regression slope, intercept, standard deviation and coefficient of variation. We used most of these target values in our benchmark experiment to assess how our pairwise calibration strategy influences the recognition of the presented LCS as NSIM devices as defined by EPA. In this study, we did not look at the standard deviation or coefficient of variation. Since we used only two LCS, our experimental setup did not fulfil the requirements to calculate both performance metrics according to EPAs test protocols.

The REU is a performance metric, which is used for the assessment of the compliance of data quality objectives (DQOs) set in the European Air Quality Directive 2008/50/EC (AQD) (Directive 2008/50/EC, 2008; Yatkin et al., 2022). The REU is used in LCS studies (Spinelle et al., 2015; Castell et al., 2017; Cordero et al., 2018; Bigi et al., 2018; Liu et al., 2019; Bagkis et al., 2021; Ratingen et al., 2021; Bagkis et al., 2022), yet it is not a common sight to describe measurement uncertainty (Karagulian et al., 2019). While LCS currently can't meet the strict requirements for reference measurements in the AQD, their measurements can at least meet less strict DQOs. For this reason, LCS can provide valuable supplemental information like indicative measurements next to regulatory fixed measurements given by air quality stations for the assessment of air quality. This is acknowledged through the recently developed European technical specifications by CEN for gas sensors and particulate matter sensors (CEN/TS 17660-1:2021, 2021; CEN/TS 17660-2:2024, 2024). Both CEN/TS present classification schemes for LCS, which respect the requirements for indicative measurements (class 1) and objective estimation (class 2) defined in the AQD Directive 2008/50/EC (2008). Furthermore, the CEN/TS offer a classification for LCS, being out of scope of the DQOs set in the AQD. Those LCS fulfil more relaxed performance criteria and provide non-regulatory measurements (class 3). For instance, LCS classified as class 3 air sensors can be applied in citizen science studies or can be used for educational purposes to raise environmental awareness. Finally, to classify the LCS as class 1, class 2 or class 3 air sensor devices, we only used the REU estimated at the air pollutant limit values (LV) in accordance with CEN/TS (CEN/TS 17660-1:2021, 2021; CEN/TS 17660-2:2024, 2024). The LVs were obtained from CEN/TS (CEN/TS 17660-1:2021, 2021; CEN/TS 17660-2:2024, 2024). The DQO of class 1, class 2 and class 3 correspond to specific relative expanded uncertainties defined in CEN/TS for each air pollutant (Tables S1 and S2). Recently, the global air quality guidelines were updated by the World Health Organization (WHO) based on the latest systematic reviews of exposure-response studies (WHO, 2021). The European Union Parliament and the European Council agreed to a new revised AQD because of this development (Directive (EU) 2024/2881, 2024). The latest revised Directive (EU) 2024/2881 aligned its standards closer to the latest WHO air quality guidelines and introduced



stricter LVs and updated DQOs for indicative measurements and objective estimation. Please note, that the presented CEN/TS might change in the future to reflect the changes in Directive (EU) 2024/2881. To visualize the REU in the statistics software
R we followed the study of Diez et al. (2022) as a reference, who made their code and data available. Furthermore, we used different smoothers (GAM, LOESS) in the REU figures depending on the sample size of the calibration data (Figs. 8, 9, 10, 11).

We calculated the REU according to the Guide for the demonstration of equivalence (GDE) following the introduced CEN/TS
(GDE, 2010; CEN/TS 17660-1:2021; CEN/TS 17660-2:2024). The REU is calculated through Eq. (1):

$$REU(y_i) = \frac{2\left(\frac{RSS}{(n-2)} - u^2(x_i) + [b_o + (b_1 - 1)x_i]^2\right)^{1/2}}{y_i} \; x \; 100, \tag{1}$$

with

$$RSS = \sum(y_i - b_0 - b_1 x_i)^2,$$


where $b_0$ is the intercept and $b_1$ the slope of the orthogonal regression of $y_i$ against $x_i$. $x_i$ are the reference measurements given through the measurement instruments of the AEMS and $y_i$ are the model calibrated LCS measurements provided by the AELCM units, which together form $n$ pairs of observation data. RSS is the residual sum of squares resulting from the orthogonal regression. $u$ describes the uncertainty of the AEMS measurement instrument, which was obtained for every AEMS
measurement instrument through CEN/TS (CEN/TS 17660-1:2021, 2021; CEN/TS 17660-2:2024, 2024).

## 3 Results and discussion

### 3.1 Air pollution and meteorological situation

The environmental conditions and pollution concentrations based on hourly means are provided in Table 2. In our work, every LCS showed the premise of being a good-quality source of information according to the employed Spearman rank correlation
Rs (Table 2). We used the hourly means of the raw output of the LCS and of the reference station AEMS to calculate Rs. Considering the observed gas concentration range in Table 2 and the CEN/TS, it can be inferred that the LV for CO (10 mg m$^{-3}$) and NO$_2$ (200 µg m$^{-3}$) were not reached at the measurement site. Thus, we could not classify the sensors according to CEN/TS 17660-1:2021. Classification according to CEN/TS 17660-1:2021 and CEN/TS 17660-2:2024 were possible for O$_3$ and PM$_{2.5}$ since the hourly LV for O$_3$ (120 µg m$^{-3}$) and the daily LV for PM$_{2.5}$ (30 µg m$^{-3}$) were reached in their respective test
periods. Considering the observed concentration ranges for each air pollutant at our urban background collocation site (Table 2), we decided to do an in-depth analysis focussing on the O$_3$ and PM$_{2.5}$ LCS in this study. Nevertheless, the analytical results for the employed CO and NO$_2$ LCS are provided in the supplement of this study since in the respective test periods the thresholds for the averaged concentrations for each air pollutant at the urban background collocation site were reached once at





least as recommended by EPA (Duvall et al., 2021a; Duvall et al., 2021b). The recommended thresholds are 1 h average concentrations of 60 ppb for $O_3$, 30 ppb for $NO_2$, and 500 ppb for CO. The recommended threshold for the 24 h average is 25 µg m$^{-3}$ for PM$_{2.5}$. The EPA suggests that these averaged concentrations must be reached at least once during a (30-day) test period (Duvall et al., 2021a; Duvall et al., 2021b).

**Table 2.** Statistics based on the hourly means of the different atmospheric variables measured by the AEMS. For the calculation of the Spearman rank correlation (Rs) all raw hourly LCS data for every individual sensor are used from AELCM009 and AELCM010. The AEMS data are used as reference for the correlation.

| Measured Variable | Timespan | Min. | 5th Percentile | 25th Percentile | Mean | 75th Percentile | 95th Percentile | Max. | Rs AELCM 009/010 |
|---|---|---|---|---|---|---|---|---|---|
| $O_3$ (ppb) | 11/01/22–11/01/23 | 0.03 | 1.07 | 12.19 | 26.43 | 37.97 | 58.42 | 81.87 | 0.75/0.68 |
| $NO_2$ (ppb) | 11/01/22–10/01/23 | 0.02 | 1.01 | 2.62 | 7.23 | 10.10 | 19.99 | 38.54 | 0.75/0.77 |
| CO (ppb) | 11/01/22–10/01/23 | 74.35 | 94.53 | 117.83 | 181.46 | 213.52 | 368.11 | 1013.46 | 0.85/0.83 |
| PM$_{2.5}$ (µg m$^{-3}$) | 11/01/22–06/01/23 | 0.14 | 1.78 | 4.41 | 9.72 | 12.92 | 24.91 | 153.22 | 0.95/0.95 |
| Temperature (°C) | 11/01/22–11/01/23 | -10.02 | -1.39 | 4.97 | 11.29 | 17.14 | 24.98 | 35.65 | 0.99/0.99 |
| Relative Humidity (%) | 11/01/22–11/01/23 | 18.69 | 35.31 | 58.24 | 71.48 | 87.29 | 92.60 | 96.33 | 0.91/0.96 |
| Pressure (hPa) | 11/01/22–11/01/23 | 937.2 | 949.1 | 958.7 | 962.5 | 966.9 | 973.8 | 983.1 | – / – |

**3.2 Baseline single training results**

We evaluated the calibration model output considering the training period and test period of each LCS targeting a specific air substance. The performance metrics in Table 3 highlight the general robustness and overall good performance of the found calibration models. All LCS models for $O_3$ and PM$_{2.5}$ for both AELCM boxes were able to reflect the patterns in the reference data well. For the $O_3$ calibration model training period $R^2$ was 0.92–1.00 and for the test period 0.93–0.98, for the PM$_{2.5}$ calibration model training period $R^2$ was 0.93–0.97 and for the test period 0.84–0.93. Considering the sensor error target by EPA (RSME ≤ 5 ppb), it was reached for every $O_3$ sensor calibration model applied to the training period (RMSE: 0.80–4.35 ppb). It was mostly reached or at least approached during the test period (RMSE: 3.62–5.84 ppb). Instead of hourly means, the recommended performance metrics and target values by EPA for PM$_{2.5}$ are based on 24 h averages (RSME ≤ 7 µg m$^{-3}$). Considering the results for the model-adjusted hourly means of the PM$_{2.5}$ air sensor output for the training period (RMSE: 1.45–2.51 µg m$^{-3}$) and the test period (RMSE: 2.04–3.02 µg m$^{-3}$), the calculated daily averages and their corresponding PM$_{2.5}$ sensor error target were fulfilled for each calibration model.



While the $O_3$ sensor calibration models based on the machine learning techniques RF and XGB performed the best in regards of $R^2$, MAE and RMSE in the training period, it is not the case in the test period. Table 3 shows the results of a single calibration using different calibration models. The tree-based algorithms represented through RF and XGB have the constraint, that they are bound by their calibration space (Bigi et al., 2018). Tree-based models can only estimate within the bounds of the calibration space, which is defined by the input data of the selected features (Bigi et al., 2018). Here, the input data is extracted

from the ST period from January to June 2022. MLR and RR don't have a constraint like tree-based models in regards of calibration space. Therefore, the more pronounced performance decline from training period to test period compared to the MLR and RR calibration approaches appear reasonable. MLR and RR calibration models seem to be an appropriate option for low-cost $O_3$ air sensors in a ST setup, because apparently there is no meaningful performance benefit in using tree-based calibration models considering the calculated performance metrics for the test period in Table 3. The same holds true for $PM_{2.5}$.

Given that a training period spans several months, MLR and RR calibration models should be used instead of tree-based models, if the goal is to calibrate the chosen $O_3$ and $PM_{2.5}$ LCS in a ST setup. Further explanations for this recommendation can be found in section 3.3.

It must be mentioned that identical LCS sensor units like the calibrated AS-B431 and SAG-SPS30 performed differently at

the same location when inspecting the calculated $R^2$, RMSE and MAE values. The raw output data given by the AS-B431 for $O_3$ (net voltages) and the SAG-SPS30 for $PM_{2.5}$ (mass concentrations) of both AELCM boxes were almost perfectly correlated ($R^2 \geq 0.97$) during the collocation period. This implies changes in sensor signals were responses to changing environmental conditions (e. g. air pollution, ambient temperature and humidity) and not related to sensor-to-sensor variability. Bittner et al. (2022) reported the same behaviour for Alphasense electrochemical gas sensors. Performance differences between the same

LCS model units after calibration are possibly related to the varying performance of the other sensors used in the LCS calibration models.

**Table 3.** Performances of LCS calibration models for $O_3$ and $PM_{2.5}$ for each AELCM box using hourly means. Results are for the $O_3$ training dataset (11 January, 19:00:00–10 June 2022, 18:00:00) and $O_3$ test dataset (10 June 2022, 19:00:00–11 January 2023, 17:00:00) as well as
for the $PM_{2.5}$ training dataset (11 January, 19:00:00–10 June 2022, 18:00:00) and $PM_{2.5}$ test dataset (10 June 2022, 19:00:00–7 January 2023, 00:00:00). MLR = Multiple Linear Regression, RR = Ridge Regression, XGB = Extreme Gradient Boosting, RF = Random Forest.

| Model target | Training $R^2$ | Training MAE (ppb) | Training RMSE (ppb) | Test $R^2$ | Test MAE (ppb) | Test RMSE (ppb) |
|---|---|---|---|---|---|---|
| $O_3$ (MLR, 009) | 0.98 | 1.57 | 1.97 | 0.98 | 2.49 | 3.62 |
| $O_3$ (MLR, 010) | 0.93 | 3.05 | 3.84 | 0.93 | 4.05 | 5.13 |
| $O_3$ (RR, 009) | 0.97 | 2.00 | 2.52 | 0.97 | 2.98 | 3.91 |
| $O_3$ (RR, 010) | 0.92 | 3.51 | 4.35 | 0.94 | 3.69 | 4.81 |
| $O_3$ (XGB, 009) | 0.99 | 0.84 | 1.07 | 0.97 | 2.97 | 3.75 |
| $O_3$ (XGB, 010) | 0.99 | 1.44 | 2.08 | 0.93 | 4.21 | 5.84 |
| $O_3$ (RF, 009) | 1.00 | 0.59 | 0.80 | 0.96 | 3.42 | 4.51 |
| $O_3$ (RF, 010) | 0.99 | 0.80 | 1.08 | 0.93 | 3.87 | 5.12 |



| | Training $R^2$ | Training MAE ($\mu g\ m^{-3}$) | Training RMSE ($\mu g\ m^{-3}$) | Test $R^2$ | Test MAE ($\mu g\ m^{-3}$) | Test RMSE ($\mu g\ m^{-3}$) |
|---|---|---|---|---|---|---|
| PM$_{2.5}$ (MLR, 009) | 0.95 | 1.22 | 1.90 | 0.92 | 1.54 | 2.69 |
| PM$_{2.5}$ (MLR, 010) | 0.96 | 1.18 | 1.85 | 0.93 | 1.13 | 2.04 |
| PM$_{2.5}$ (RR, 009) | 0.93 | 1.76 | 2.48 | 0.89 | 1.82 | 2.63 |
| PM$_{2.5}$ (RR, 010) | 0.94 | 1.58 | 2.27 | 0.91 | 1.38 | 2.04 |
| PM$_{2.5}$ (XGB, 009) | 0.94 | 1.46 | 2.29 | 0.84 | 1.55 | 2.97 |
| PM$_{2.5}$ (XGB, 010) | 0.95 | 1.37 | 2.51 | 0.85 | 1.39 | 3.02 |
| PM$_{2.5}$ (RF, 009) | 0.97 | 0.97 | 1.56 | 0.87 | 1.37 | 2.85 |
| PM$_{2.5}$ (RF, 010) | 0.97 | 0.90 | 1.45 | 0.89 | 1.06 | 2.35 |

## 3.3 Extended training results and EPA performance targets

To assess seasonal differences in air sensor performance we calculated the suggested performance metrics by EPA on a 30-
day basis, namely the RMSE (error), $R^2$ (linearity), slope (bias) and intercept (bias). The EPA also provided target values for
each of these performance metrics, which are highlighted in red in the circular bar plots (e.g. Fig. 2). We used the absolute
value of the calculated intercept and the difference between the calculated model slope and the ideal slope of 1 for each
calibration model to improve the interpretability of the figures. The original intercepts and slopes can be found in the
supplement (Tables S29–S52). Circular bar plots are a visual tool to evaluate if an ET approach is beneficial in regards of
enhancing the qualitative and quantitative validity of calibrated LCS output compared to a ST approach. In addition, they
enhance the visual distinction between the different calibration techniques, i.e. MLR, RR, and the machine learning algorithms
RF and XGB.

For the most part, O$_3$ sensor calibration model performance benefitted from an extended training. The performance gains
highly varied in magnitude dependent on the performance metric (Intercept, slope, RMSE, $R^2$), ET length (1 month to 3
months) and calibration model (MLR, RR, RF and XGB). For ETs of 1 month, 2 months and 3 months and for each calibration
model both calibrated O$_3$ sensors correlated quite well with the hourly reference data during summer, autumn and winter,
which is reflected through the coefficient of determination ($R^2$ (ST): 0.79–0.98; $R^2$ (ET): 0.86–0.98). Only once the target
value range for $R^2$ was missed, which was for AELCM010 and the RF calibration model in TP5 for the ST variant. But an
extended training resulted in reaching the target value range for $R^2$ in TP5 for this calibration model. To summarize, a single
training period of 5 months was almost sufficient to reach the target value range for $R^2$ ($R^2 \geq 0.80$) for every TP and every O$_3$
sensor calibration model. High $R^2$ values for the calibrated O$_3$ sensor units of the same type (AS-B431) for periods associated
with northern hemisphere winter and warmer months ("ozone season") are in agreement with other low-cost air sensor studies
(Zimmerman et al., 2018; Zauli-Sajani et al., 2021). Considering the performance metric $R^2$, it can be noted that MLR and RR
competed well against the machine learning techniques RF and XGB during all ET configurations.



We found a distinct difference in gas sensor performance for $R^2$ between warmer periods and colder periods considering the employed $NO_2$ and CO sensors, which implied the existence of limiting factors in sensor calibration. Generally, TP4 ($\approx$ September) was the first test period, where $NO_2$ sensor calibration models and CO sensor calibration models entered the $R^2$ target range for $NO_2$ sensors ($R^2 \geq 0.70$) and CO sensors ($R^2 \geq 0.80$), which are recommended by EPA (Figs. S5, S6, S7, S12, S13, S14). In the following months (TPs), $NO_2$ and CO sensor calibration models were available, which performed in the boundaries of their targeted $R^2$ range. We assume for TP1 till TP3, that an interplay between environmental interferences and limited sensor sensitivity at lower ambient concentrations of $NO_2$ and CO played a crucial role for the overall low sensor performances in those warmer periods, considering the findings in other studies (Cross et al., 2017; Hagan et al., 2018). The mean reference values for $NO_2$, CO, air temperature and relative humidity for every TP can be found in the supplement (Figs. S3 and S4). In the warmer periods MLR and RR LCS calibration models performed notably worse for $NO_2$ sensors, which is expressed through $R^2$. The reason could be, that during these periods non-linearity was introduced to the sensor signals due to increased air temperatures at low concentrations (Cross et al., 2017; Hagan et al., 2018), where non-linear models like RF and XGB outperformed linear models like MLR and RR.

It must be said that an extension of the training period for the $O_3$ and $PM_{2.5}$ calibration models had overall only a small impact on $R^2$, when comparing the LCS calibration models between their ST and ET variants (Figs. 2, 3, 4, 5, 6, 7). $R^2$ values only occasionally experienced stronger positive changes between at least 0.05 and 0.09 through extended training for some TPs and mainly for the $O_3$ LCS and the RF and XGB calibration models (Tables S29–S52). The correlative performance of the $O_3$ and $PM_{2.5}$ calibration models for ST were already quite high. The calibrated $PM_{2.5}$ sensors correlated quite well with the daily reference data during summer, autumn and winter ($R^2$ (ST): 0.76–0.99; $R^2$ (ET): 0.79–0.99) for ETs of 1 month, 2 months and 3 months and for each calibration model. A ST period of 5 months was sufficient to reach the target value range for $R^2$ ($R^2 \geq 0.70$) for every TP and every $PM_{2.5}$ sensor calibration model. High $R^2$ values for the calibrated $PM_{2.5}$ sensor units (SAG-SPS30) for periods associated with northern hemisphere winter (heating season) and warmer months are in agreement with other low-cost air sensor studies, where the same sensor type was factory-calibrated or model-calibrated (Vogt et al., 2021; Gäbel et al., 2022; Shittu et al., 2025). Distinctive benefits for applying an extended training to calibration models were rather identified for performance metrics, which describe the bias and error.

Using ST, our LCS calibration models were trained on data between January and June. The test periods TP1 till TP3 ($\approx$June–September) in Fig. 2, Fig. 3 and Fig. 4 are the most relevant TPs for the assessment of the performance (Intercept, slope, RMSE) of our $O_3$ sensor calibration models, because of the elevated ozone concentrations and the health relevancy of ozone in these periods in the northern hemisphere (Hertig et al., 2019; Jahn and Hertig, 2021). TP1 is the only period, where we can see a change in performance of LCS calibration models for a single $O_3$ sensor (AELCM010) dependent on ET length over all introduced ET lengths (ETs of 1, 2 and 3 months).




**Figure 2.** Performance metrics of the single O₃ LCS in each AELCM box, calculated from hourly mean values after calibration. Metrics are presented for each calibration model, test period (TP), and calibration variant (Single training (ST) and extended training (ET)). Models are ordered by performance from highest to lowest in each period. The extended training is characterized by the one-month variant for each AELCM box. Values highlighted in red describe the least accepted target value given by EPA for each performance metric (|Intercept| (a), RMSE (b), ΔSlope (c), R² (d)).







**Figure 3.** Performance metrics of the single O₃ LCS in each AELCM box, calculated from hourly mean values after calibration. Metrics are presented for each calibration model, test period (TP), and calibration variant (Single training (ST) and extended training (ET)). Models are ordered by performance from highest to lowest in each period. The extended training is characterized by the two-month variant for each AELCM box. Values highlighted in red describe the least accepted target value given by EPA for each performance metric (|Intercept| (a), RMSE (b), ΔSlope (c), R² (d)).





**Figure 4.** Performance metrics of the single O$_3$ LCS in each AELCM box, calculated from hourly mean values after calibration. Metrics are presented for each calibration model, test period (TP), and calibration variant (Single training (ST) and extended training (ET)). Models are ordered by performance from highest to lowest in each period. The extended training is characterized by the three-month variant for each AELCM box. Values highlighted in red describe the least accepted target value given by EPA for each performance metric (|Intercept| (a), RMSE (b), ΔSlope (c), R$^2$ (d)).



Overall, the bias worsens with ET lengths of 2 months and 3 months. The most pronounced degradation of intercept and slope

can be seen for RF and XGB, due to reducing the amount of summer training data provided by AELCM010, which decreases the relevant calibration space for the XGB and RF calibration models. While the XGB calibration model with ETs of 1 month almost reached the intercept target value range (|Intercept| ≤ 5 ppb) in TP1, not a single calibration model even approached the target value range with other ET lengths. The RF and XGB calibration models using only ST and ETs of 3 months were outside the slope target value range ($\Delta$Slope ≤ 0.2) in TP1. The decrease in performance was also expressed by a decrease of the

number of calibration models, which were within the target value range for the RMSE (RMSE ≤ 5 ppb). Most calibration models did not achieve the target RMSE range during TP1. The maximum was two models, both using ET lengths of 1 month (XGB and RF). Considering all calculated performance metrics from Fig. 2, Fig. 3 and Fig. 4 in TP1, the XGB calibration model with ETs of 1 month was almost able to reach all target values provided by EPA. Looking at TP1 and TP3 with respect to bias and error, in general XGB and RF calibration models suffered the most under a lack of training data (ST mode) and a

loss of summer training data due to longer ETs, which reflect no or a reduced recalibration cycle, respectively. Comparing MLR and RR calibration models with XGB and RF calibration models for AELCM010 regarding bias and error in TP1, TP2 and TP3, it becomes evident that the application of extended training to the machine learning techniques can yield substantial improvements, whereas the absence of such training may result in markedly higher bias and error. In these periods the impact on bias reduction and error reduction due to an ET is more pronounced for the RF and XGB calibration models related to the

$O_3$ sensor employed with AELCM010. Considering our experimental setup at an urban background site as well as the calculated bias and error metrics for TP1 to TP7, we conclude the following for LCS air pollution studies that aim to make quantitative statements about $O_3$ employing AS-B431 sensor units: 1) MLR and RR calibration models should be employed when extended training is not an option, but a single multi-month training period is available, which accounts for seasonal variations in atmospheric conditions (meteorological and air pollution factors) and thus a wide range of environmental

influences on the sensor signal. 2) If extended training is an option in the form of monthly recalibration, RF and XGB calibration models appear to be the more sensible option.

Unlike $O_3$, all test periods in Fig. 5, Fig. 6 and Fig. 7 were relevant for assessing the performance of our $PM_{2.5}$ sensor calibration models from a health perspective, as unhealthy levels of $PM_{2.5}$ can be present throughout the year due to the diverse sources

of ambient $PM_{2.5}$. Main anthropogenic sources include industrial emissions, ground transport emissions, biomass burning and the secondary formation of fine particulate matter classified as $PM_{2.5}$ (Thunis et al., 2021; Gu et al., 2023; Chowdhury et al., 2023; Zauli-Sajani et al., 2024). Natural sources of fine particles include wildfires (Chowdhury et al., 2024) and dust events, such as Saharan dust transported to different latitudes (Karanasiou et al., 2012). Generally, weather conditions and the atmospheric state influence the transport, mixing ratio, transformation and deposition of air substances; hence they are

important factors defining the air quality level (Dayan and Levy, 2002; Russo et al., 2014; Russo et al., 2016; Reizer and Juda-Rezler, 2016).





A MLR calibration model was the only one that satisfied every recommendation by EPA for PM$_{2.5}$ sensor bias (|Intercept| ≤ 5
µg m$^{-3}$; ΔSlope ≤ 0.35) in all TPs, which is shown in Fig. 5, Fig. 6 and Fig. 7. Here, the MLR calibration model with ET
reached the target range for the slope in TP1, which the other calibration models did not. The intercept target range was met
by all calibration models with ST in each TP. No extended training was needed here. The same applied for the PM$_{2.5}$ sensor
error target range (RSME ≤ 7 µg m$^{-3}$). Considering how often a MLR calibration model was the best performing model in
regards of sensor bias and sensor error, we conclude that a MLR calibration model is sufficient to improve the quantitative
validity of raw SAG-SPS30 data. RF and XGB did not offer a substantial alternative, visible by their performance metrics.
This is emphasized for instance in Fig. 6, where the best-performing machine learning model, RF with ET, barely offers more
than a small performance improvement compared to a MLR calibration model with ST. Looking at all ST calibration models
and ET calibration models, there is generally very little change in quantitative performance following an extended training
approach. In our collocation experiment, the chosen ST period appears to be sufficient to train robust calibration models, which
perform well in the following test periods in an urban background setting. Therefore, recalibration appears largely unnecessary
for the SAG-SPS30 when considering only the EPA performance targets discussed in this section, rather than the more stringent
DQOs outlined in section 3.4. This is particularly notable given that both the RF and MLR calibration models, trained using
the ST period, nearly met the slope target range in TP1.

A calibrated SAG-SPS30 performed usually well in all performance categories in each TP. This could be due to the raw sensor
data quality, which may be influenced by how the measurement principle was technically integrated into the SAG-SPS30, or
because of the out-of-the-box calibration algorithm provided by Sensirion (Vogt et al., 2021), from which our calibration
models may have benefited. To satisfy the EPA performance recommendations through LCS calibration was less challenging
for a SAG-SPS30 compared to the Alphasense electrochemical gas sensors. An extended training is recommended to achieve
the best possible sensor performance for the Alphasense electrochemical gas sensors. Here, an ET length of 1 month seemed
to be the overall most sensible choice (Figs. 2, 3, 4, Figs. S5, S6, S7, Figs. S12, S13, S14, Tables S5-S40), likely because gas
sensor performance is influenced by local atmospheric conditions, which experience seasonal variation.

Therefore, more frequent pairwise recalibrations are expected to improve the calibration process. Our performance results
implied, that the electrochemical LCS AS-B431, AS-B43F and AS-B4 benefit the most from a pairwise recalibration every 30
days, where the pairwise calibration gets extended by another 30 days. With two AELCM units in this mode, we can recalibrate
during the same season while continuously collecting in situ data. This allows us to account for changing environmental
conditions and their influence on gas sensor performance. Our results next to other studies show, that the likelihood of well
performing calibration models for the employed low-cost air sensors increases with a sufficient amount of training data (Zauli-
Sajani et al., 2021; Nowack et al., 2021) and with raw LCS measurement data, which are not dominated by noise due to low
concentrations because of sensor sensitivity limits (Zimmerman et al., 2018).





**Figure 5.** Performance metrics of the single PM$_{2.5}$ LCS in each AELCM box, calculated from daily mean values after calibration. Metrics are presented for each calibration model, test period (TP), and calibration variant (Single training (ST) and extended training (ET)). Models are ordered by performance from highest to lowest in each period. The extended training is characterized by the one-month variant for each AELCM box. Values highlighted in red describe the least accepted target value given by EPA for each performance metric (|Intercept| (a), RMSE (b), ΔSlope (c), R$^2$ (d)).




**Figure 6.** Performance metrics of the single PM$_{2.5}$ LCS in each AELCM box, calculated from daily mean values after calibration. Metrics are presented for each calibration model, test period (TP), and calibration variant (Single training (ST) and extended training (ET)). Models are ordered by performance from highest to lowest in each period. The extended training is characterized by the two-month variant for each AELCM box. Values highlighted in red describe the least accepted target value given by EPA for each performance metric (|Intercept| (a), RMSE (b), ΔSlope (c), $R^2$ (d)).






**Figure 7.** Performance metrics of the single PM$_{2.5}$ LCS in each AELCM box, calculated from daily mean values after calibration. Metrics are presented for each calibration model, test period (TP), and calibration variant (Single training (ST) and extended training (ET)). Models are ordered by performance from highest to lowest in each period. The extended training is characterized by the three-month variant for each AELCM box. Values highlighted in red describe the least accepted target value given by EPA for each performance metric (|Intercept| (a),
RMSE (b), ΔSlope (c), R$^2$ (d)).



## 3.4 Extended training results and data quality objectives

The EU Air Quality Directive 2008/50/EC (2008) and the new Directive (EU) 2024/2881 (2024) provide DQOs for regulatory grade measurement devices, which LCS are not. But LCS have a legitimate role alongside those regulatory grade monitoring systems as air sensors for indicative measurements and objective estimations. We applied REU plots to analyse the possible end-use applications of the employed calibrated AELCM sensors, considering the DQOs and LVs for air sensor classification provided by the CEN/TS. The DQOs used in the sensor test protocols CEN/TS 17660-1:2021 and CEN/TS 17660-2:2024 are based on AQD Directive 2008/50/EC (2008). REU plots helped to describe the measurement uncertainty "point by point" of the calibrated LCS, complementing the use of single-value error metrics (global performance metrics) applied in Sect. 3.2 and Sect. 3.3 (Diez et al., 2024). They provide deeper insight into the error structures and information content of calibrated LCS data (Diez et al., 2022).

Figures 8 and 9 show the "point by point" measurement uncertainty of the "classical" MLR $O_3$ calibration model and the machine learning-based RF $O_3$ calibration model. The fluctuation in measurement uncertainty across the observed range was greater for the calibrated $O_3$ LCS data of AELCM010, which is shown in the top rows of the REU plots. The calibrated $O_3$ LCS data of AELCM009 and AELCM010 with ST meet the class 1 DQO (REU ≤ 30 %), but the calibrated data of AELCM009 reached it more reliably even at lower measured concentrations. The REU values at the $O_3$ LV of 120 µg m$^{-3}$ imply, that both calibrated $O_3$ LCS can be classified as class 1 sensor systems and therefore can be used for indicative measurements. It must be said that we didn't follow all activities and principles, which are relevant for the classification according to CEN/TS 17660-1:2021 (2021) and CEN/TS 17660-2:2024 (2024). This includes laboratory tests, which were not part of this study.

As in Sect. 3.2, performance differences between identical LCS sensor units are evident once more, but visually detectable across the entire observed concentration range of ambient $O_3$, an aspect global performance metrics (e.g. RMSE, $R^2$, MAE) fail to capture (Diez et al., 2022). The top rows in both figures (also Figs. S19 and S20) depict a differing response of the employed calibrated sensor units to the same environmental conditions experienced at the station site during the collocation period. Possible reasons for these differences in sensor behaviour were explained in Sect. 3.2. Extending the calibration model training period and therefore expanding the calibration space is advised for machine learning methods, as evidenced by the REU plots in Fig. 9 and Fig. S19. In the 3 months ET AELCM010 was active in TP1 to TP3, the time when the highest $O_3$ concentrations were observed (Fig. S1). In the 2 months and 1 month ET AELCM009 was active in TP3 and TP2, in that order. The lack of further summer training data in the 3 months ET resulted visibly in increased REU values above 100 µg m$^{-3}$ (Fig. 9, bottom left) for AELCM010. The other two ET lengths provide further summer training data to the RF calibration models for the $O_3$ LCS belonging to AELCM009 and AELCM010. This resulted in a reduced measurement uncertainty for higher concentrations in TP1 until TP3 (Fig. 9, bottom middle and bottom right), being not the case for both RF calibrated LCS using





only ST (Fig. 9, top row). If pairwise calibration is considered for a LCS measurement campaign, we recommend using two calibrated LCS meeting the same DQO, in order to ensure consistent in situ data quality as demonstrated in Fig. 8 and Fig. 9.


Figures 10 and 11 show the "point by point" measurement uncertainty for the $PM_{2.5}$ calibration models based on MLR and RF. The MLR and RF calibrated $PM_{2.5}$ datasets of AELCM009 showed greater measurement uncertainty across the observed daily means of $PM_{2.5}$. With the ST calibrated datasets, REU values met the Class 1 DQO (REU $\leq$ 50 %) less consistently compared to the REU values related to calibrated $PM_{2.5}$ data of AELCM010. The REU values at the $PM_{2.5}$ LV of 30 µg m$^{-3}$

using ST imply, that the MLR calibrated $PM_{2.5}$ LCS of AELCM009 can be classified as a class 2 sensor system for objective estimation (REU $\leq$ 100 %), whereas the MLR calibrated $PM_{2.5}$ LCS of AELCM010 can be classified as a class 1 sensor system for indicative measurements. RF calibration models suggest that both $PM_{2.5}$ LCS accomplish the highest tier of sensor systems (class 1), achieving indicative measurements at the $PM_{2.5}$ LV. Above 5 µg m$^{-3}$ both RF calibrated $PM_{2.5}$ LCS show (almost) consistently data meeting the class 1 DQO for the ST mode. The non-aligning patterns in relative error between the ST

calibrated SAG-SPS30 units indicate that the employed calibrated sensor units respond differently under identical environmental conditions (Figs. 10, 11, Figs. S21, S22), as previously observed with the AS-B431 units measuring $O_3$.

ET for the MLR and RF calibration models helped build continuous LCS time series that more consistently meet the class 1 DQO, using both calibrated SAG-SPS30 units (Figs. 10, 11, bottom row). ET to achieve more consistency in data quality was

especially relevant for the $PM_{2.5}$ LCS employed with AELCM009. Figure 11 shows, that an extended training characterized by the one-month variant was the most beneficial to reduce measurement uncertainty for higher concentrations of $PM_{2.5}$. We conclude that higher sensor system tiers for LCS can be achieved through ET, thereby broadening the scope of applications for a LCS.






**Figure 8.** Calculated REU values for MLR calibrated O₃ LCS hourly data belonging to the test periods (TP1–TP7, 10 June 2022–11 January 2023) of AELCM009 and AELCM010. The calibration variants are single training (ST) (top row, left: AELCM010, right: AELCM009) and extended training (ET) (bottom row). The extended training is characterized by ET variants of 1, 2 and 3 months for each AELCM box. Horizontal dashed lines describe the data quality objectives (O₃ Class 1 DQO = 30 %, Class 2 DQO = 75 % and Class 3 DQO = 200 %).

The vertical dashed line describes the limit value for O₃ (LV = 120 µg m⁻³). The fitted smooth curve (red) is based on a generalized additive model (GAM). Data density is shown through colour, where darker colours express lower data density and brighter colours express higher data density.







**Figure 9.** Calculated REU values for RF calibrated $O_3$ LCS hourly data belonging to the test periods (TP1–TP7, 10 June 2022–11 January 2023) of AELCM009 and AELCM010. The calibration variants are single training (ST) (top row, left: AELCM010, right: AELCM009) and extended training (ET) (bottom row). The extended training is characterized by ET variants of 1, 2 and 3 months for each AELCM box. Horizontal dashed lines describe the data quality objectives ($O_3$ Class 1 DQO = 30 %, Class 2 DQO = 75 % and Class 3 DQO = 200 %). The vertical dashed line describes the limit value for $O_3$ (LV = 120 µg m$^{-3}$). The fitted smooth curve (red) is based on a generalized additive model (GAM). Data density is shown through colour, where darker colours express lower data density and brighter colours express higher data density.







**Figure 10.** Calculated REU values for MLR calibrated PM$_{2.5}$ LCS daily data belonging to the test periods (TP1–TP7, 11 June 2022–6 January 2023) of AELCM009 and AELCM010. The calibration variants are single training (ST) (top row, left: AELCM010, right: AELCM009) and extended training (ET) (bottom row). The extended training is characterized by ET variants of 1, 2 and 3 months for each AELCM box. Horizontal dashed lines describe the data quality objectives (PM$_{2.5}$ Class 1 DQO = 50 %, Class 2 DQO = 100 % and Class 3 DQO = 200 %). The vertical dashed line describes the limit value for PM$_{2.5}$ (LV = 30 µg m$^{-3}$). The fitted smooth curve (red) is based on locally estimated scatterplot smoothing (LOESS). Data density is shown through colour, where darker colours express lower data density and brighter colours express higher data density.





**Figure 11.** Calculated REU values for RF calibrated PM$_{2.5}$ LCS daily data belonging to the test periods (TP1–TP7, 11 June 2022–6 January 2023) of AELCM009 and AELCM010. The calibration variants are single training (ST) (top row, left: AELCM010, right: AELCM009) and extended training (ET) (bottom row). The extended training is characterized by ET variants of 1, 2 and 3 months for each AELCM box. Horizontal dashed lines describe the data quality objectives (PM$_{2.5}$ Class 1 DQO = 50 %, Class 2 DQO = 100 % and Class 3 DQO = 200 %). The vertical dashed line describes the limit value for PM$_{2.5}$ (LV = 30 μg m$^{-3}$). The fitted smooth curve (red) is based on locally estimated scatterplot smoothing (LOESS). Data density is shown through colour, where darker colours express lower data density and brighter colours express higher data density.




**3.5 Implications for sustainable LCS networks and future outlook**

Our concept for an effective, sustainable and manageable LCS network involves the question, who is the main target population in regards of health protection from environmental exposure such as air pollution and heat considering the advantages and

disadvantages of current LCS technology. We consider the most vulnerable people as our main target group, for instance children, elderly people, outdoor workers or people with pre-existing health conditions. LCS measurements can thus be placed at locations with a high density of vulnerable populations, such as retirement homes, schools, kindergartens, or outdoor workplaces. Therefore, we recommend focusing on the characteristics of the measurement scope rather than simply building a spatially dense LCS observation network. Reducing the amount of LCS and efficiently placing them by figuring out at-risk

population hotspots could reduce the management effort using a pairwise calibration strategy similar to the one we introduced in this study. Another benefit of following our calibration strategy could be an improved error minimization of LCS data resulting in higher LCS data quality compared to using complex in situ calibration strategies for error reduction and data quality assurance, considering that we are placing LCS devices right next to a station for (re-)calibration. Following our concept, a continuous data collection in situ can be achieved by using a pair of regularly maintained LCS for the same location.


Analysing whether LCS data fit their intended purpose and continue to provide viable information for the end-use application over time remains a challenge, especially in the context of a long-term measurement campaign. Stricter DQOs for regulatory grade air measurement instruments as a result of the recently updated WHO global air quality guidelines (WHO, 2021) could indirectly limit the scope of end-use applications for LCS, considering that for instance CEN/TS 17660-1:2021 (2021) and

CEN/TS 17660-2:2024 (2024) rely on Directive 2008/50/EC (2008). Both CEN/TS help to define the possible end-use applications of sensor systems. Considering the relationship between the introduced CEN/TS and the Directive 2008/50/EC, an update of both CEN/TS due to the recently published Directive (EU) 2024/2881 (2024) is not unlikely. Sensor manufacturers are called upon to consult state-of-the-art scientific literature of the air sensor research community to accelerate technological advancement while the air sensor community is called upon to rethink how LCS networks are built and managed. The latter is

important to ensure that LCS networks move beyond the status of test applications and gain recognition as long-term supplemental monitoring systems (Carotenuto et al., 2023), integrated into official networks and capable of benefitting the most vulnerable people of society.

**4 Conclusions**

We have investigated in detail how commonly used air sensors, AS-B431 for $O_3$ and SAG-SPS30 for $PM_{2.5}$, should be

recalibrated in an effort to move beyond rule-of-thumb estimations (Schmitz et al., 2021). To achieve this, we conducted a yearlong collocation campaign at an urban background station for air pollution exposure and meteorological measurements next to the University Hospital Augsburg, Germany. LCS were collocated with regulatory grade air measurement instruments and were exposed to a wide range of environmental conditions, with air temperatures between -10 and 36 °C, relative air





humidity between 19 and 96 % and air pressure between 937 and 983 hPa. The ambient concentration ranges were up to 83
ppb for $O_3$ and 153 µg m$^{-3}$ for $PM_{2.5}$. Sensor calibration models were built using linear regression techniques (Multiple Linear
Regression and Ridge Regression) and machine learning (Random Forest and Extreme Gradient Boosting). Our
methodological approach followed a pairwise calibration strategy.

We aimed to assess the benefit of expanding the calibration space of these calibration models in terms of improving the
applicability of low-cost air sensors for tasks that support regulatory monitoring, particularly with respect to meeting relevant
EPA performance targets (e.g. $R^2$, RMSE, slope, intercept) and less strict DQOs (REU; indicative measurements, objective
estimation). The performance targets and the associated classifications were provided by state-of-the-art test protocols for air
sensors by the United States Environmental Protection Agency (EPA) and the European Committee for Standardization (CEN)
and acted as a guideline for the assessment. The calibration space was expanded by extending the training period (ET) of the
calibration models using 1 up to 3 months ET periods. The different ET lengths implied whether a calibration model should
be recalibrated after a month or every 2 or 3 months. As a reference we used a shorter single training period (ST) of 5 months
to train the calibration models.

The employed $O_3$ and $PM_{2.5}$ LCS capture the temporal variations in observed $O_3$ concentrations and $PM_{2.5}$ concentrations well.
We found that the time series correlations of the $O_3$ and $PM_{2.5}$ calibration models using only ST were already quite high for
each test period (TP1–TP7) and that an extension of the training period for the $O_3$ and $PM_{2.5}$ calibration models yielded overall
only a small improvement. Linear models like MLR and RR showed a similar performance to RF and XGB. Therefore, linear
models are sufficient if favourable conditions are met and if a study only needs to capture the temporal variability. In our study,
these favourable conditions most likely refer to sensor signals that are not significantly influenced by non-linearity, as well as
observed concentration ranges where the corresponding LCS data is not dominated by noise. For gas sensors, this noise
typically arises from the limited sensitivity of LCS at lower concentrations.

Distinctive benefits for calibration models using ET were rather identified for performance metrics that target the correctness
of the absolute values. Our findings suggest that for quantitative studies, during periods characterized by elevated ground level
ozone concentrations (ozone season), recalibration is advisable after each month of $O_3$ LCS operation. In particular, the
machine learning techniques RF and XGB benefited from the increased amount of summer training data resulting from monthly
recalibrations. We showed, that MLR and RR calibration models should be employed when ET is not an option, but a single
multi-month training period is available, which accounts for seasonal variations in atmospheric conditions (meteorological and
air pollution factors). If extended training via monthly recalibration is feasible, RF and XGB calibration models appear to be
the more sensible choice, as their quantitative performance aligns particularly well with EPA guidelines for non-regulatory
supplemental and informational monitoring devices targeting $O_3$.



All test periods were relevant for assessing the quantitative strength of the introduced $PM_{2.5}$ sensor calibration models from a health perspective, as unhealthy levels of $PM_{2.5}$ can be present throughout the year due to the diverse sources of ambient $PM_{2.5}$.

A MLR calibration model using ET was the only calibration model that met all EPA-recommended performance metric goals for assessing the quantitative strength of $PM_{2.5}$ LCS data. Machine learning techniques did not offer a substantial alternative to MLR calibration. When considering only the performance metrics recommended by EPA for assessing sensor bias and error, very little change in quantitative performance was seen following an ET approach. This suggests that a ST may be sufficient for developing robust and well-performing $PM_{2.5}$ LCS calibration models. Therefore, assessing the measurement uncertainty,

as expressed by the REU, contributed to a more comprehensive evaluation of the usefulness of recalibrating the SAG-SPS30 units.

REU values helped to describe the measurement uncertainty "point by point" of the pairwise calibrated LCS, complementing the use of global performance metrics in our study. The calibrated $O_3$ LCS and $PM_{2.5}$ LCS were able to meet the class 1 DQO

for different calibration models and therefore can provide indicative measurements. The REU values suggest that extended training of the employed calibration models enables the generation of a continuous LCS time series from two identical sensor model units, more consistently meeting a targeted DQO (e.g. indicative measurements). This approach also contributes to reduced measurement uncertainty, which becomes visually noticeable as a pollutant concentration increases. Again, extending the calibration model training period and therefore expanding the calibration space is especially advised for machine learning

methods to reduce the LCS measurement uncertainty.

We conclude that achieving the highest possible quantitative validity for low-cost air sensors requires regular in-season recalibration using high-quality reference data. The response of the sensor units to changing environmental conditions at the station site, along with improved performance resulting from regular recalibration that aligns sensor output more closely with

EPA and CEN recommendations, highlights how important regular sensor maintenance is to enhance their applicability. These findings underscore the importance of rigorous data quality assurance and control for studies or monitoring networks that aim to make quantitative assertions. In general, quality assurance and quality control (QA/QC) are essential for both qualitative and quantitative research involving low-cost sensors.

A monthly recalibration using a pairwise calibration strategy with two LCS devices of the same model for in situ measurements could be a feasible approach. It is particularly valuable and resource-efficient if the aim is not to just build a spatially dense sensor network, but instead to focus on targeted, good-quality monitoring at locations with high densities of vulnerable populations, such as retirement homes, schools or kindergartens. However, this approach requires continuous access to high-quality reference equipment, which represents a major obstacle for pairwise calibration strategies such as the one

employed in this study. Continued cooperation between authorities, who provide the necessary infrastructure for calibration, and researchers, who have expertise in air sensors, is essential, especially in the view of the updated WHO global air quality

guidelines and the new Directive (EU) 2024/2881, which have respectively recommended and implemented stricter limits for major air pollutants. Focused and thoughtful supplemental and informational monitoring with the aid of well-maintained low-cost air sensors within official air quality monitoring networks increases the likelihood of realizing better air quality for the population.

## Data availability

The data of this study are available from the authors upon request.

## Author contributions

Conceptualization, P.G. and E.H.; data curation, P.G.; formal analysis, P.G.; investigation, P.G.; methodology, P.G. and E.H.; project administration, P.G; resources, E.H.; software, P.G.; supervision, E.H.; validation, P.G.; visualization, P.G.; writing—original draft preparation, P.G.; writing—review and editing, E.H. All authors have read and agreed to the published version of the manuscript.

## Competing interests

The authors declare that they have no conflict of interest.

## Acknowledgements

We thank our student assistant Nicolas Hahn (University of Augsburg, Institute for Geography) for his contribution in formatting and editing the tables and figures presented in this work. Furthermore, we thank Nicolas Hahn for his contribution in exporting the performance metrics to the presented tables in the supplement using R and Python. Nicolas Hahn extracted the mean reference values from the provided code of Paul Gäbel and created the figures S1, S2, S3 and S4 in the supplement. We used AI tools to improve the language of the published version of the manuscript.

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
