# Peer review of "Recalibration of low-cost O3 and PM2.5 sensors: Linking practices to recent air sensor test protocols"

_EGUsphere, 2025_

## Author Comment (AC1)

**Response to comments from Referee #1**

**Citation**: https://doi.org/10.5194/egusphere-2025-2677-RC1

We would like to thank you for taking the time to review our manuscript and provide valuable feedback. Our responses and proposed revisions, which we believe enhance the quality of the paper, are presented below. The comments from Referee #1 are provided in black, our responses appear in brown, and the revised or newly added text in the manuscript is shown in *italics*.

First of all, I would like congratulate the authors for the work carried out and presented in this paper. After having read the full document, I'm not sure that the conclusion or the study really answer the question asked in the title. In fact, the author ask the question of the need of re-calibration of low-cost senors but they do not really answer it in the document as the present an interesting use of sensor for ambient air monitoring ("pairwise calibration strategy") based on a monthly exchange of LCS between a collocation site and a measurement site. This strategy, somehow interesting when looking at the sensors performances is much more time consuming than a classic network installation as, at the end, 2 LCS are always running adding the necessity of installation/removal every month. However, the interesting comparison of calibration results using several training length against both US-EPA and European standards brings a lot of valuable information.

In recent years, multiple recognized organizations such as the EPA and CEN have released state-of-the-art test protocols for air sensors. These are important and much-needed tools that help to communicate the possible end-use applications of low-cost sensors to the public after their evaluation. In this work, these test protocols provide guidance for evaluating and contextualizing the actual impact of different training lengths (extended training (ET)) compared to a shorter training period (single training (ST)) on sensor performance. However, the main research question is if and how recalibration must be designed to maximize performance of the sensors.

The conclusions section (Sect. 4) of this work offers the following statements related to the question asked in the title (Recalibration of low-cost air pollution sensors: Is it worth it?) of this study:

1. Our findings suggest that for quantitative studies, during periods characterized by elevated ground level ozone concentrations (ozone season), recalibration is advisable after each month of $O_3$ LCS operation. In particular, the machine learning techniques RF and XGB benefited from the increased amount of summer training data resulting from monthly recalibrations.

2. If extended training via monthly recalibration is feasible, RF and XGB calibration models appear to be the more sensible choice, as their quantitative performance

aligns particularly well with EPA guidelines for non-regulatory supplemental and informational monitoring devices targeting $O_3$.

3. A MLR calibration model using ET was the only calibration model that met all EPA-recommended performance metric goals for assessing the quantitative strength of $PM_{2.5}$ LCS data.

4. The REU values suggest that extended training of the employed calibration models enables the generation of a continuous LCS time series from two identical sensor model units, more consistently meeting a targeted DQO (e.g. indicative measurements). This approach also contributes to reduced measurement uncertainty, which becomes visually noticeable as a pollutant concentration increases. Again, extending the calibration model training period and therefore expanding the calibration space is especially advised for machine learning methods to reduce the LCS measurement uncertainty.

5. We conclude that achieving the highest possible quantitative validity for low-cost air sensors requires regular in-season recalibration using high-quality reference data. The response of the sensor units to changing environmental conditions at the station site, along with improved performance resulting from regular recalibration that aligns sensor output more closely with EPA and CEN recommendations, highlights how important regular sensor maintenance is to enhance their applicability.

We understand the reviewer's point that the current title may not fully reflect the content, which could be expected given its provocative nature. If the title seems too strong, we propose the following possible revisions:

*Recalibration of low-cost air pollution sensors: Linking practices to state-of-the-art test protocols*

*Recalibration of low-cost air pollution sensors: Connecting calibration practices with modern test protocols*

*Recalibration of low-cost air pollution sensors for advanced performance*

Furthermore, we agree that a pairwise calibration strategy is more time-consuming than a classic network installation, particularly when sensors are installed and removed monthly in a large-scale network. However, considering our observed sensor performances, we see value and the possibility in applying a pairwise calibration strategy in small networks, especially when LCS measurement systems are deployed at locations with high densities of vulnerable populations, such as retirement homes, schools, kindergartens, or outdoor workplaces. Implementing multiple smaller-scale

LCS networks by various groups with access to adequate infrastructure for sensor calibration (e.g. research institutions, state organizations), focused on at-risk population hotspots, could help LCS realize their potential and, in fact, gain recognition as long-term supplemental monitoring systems, integrated into official networks to serve the most vulnerable people of society.

I also made some minor comment along the document listed below:

➢ Line 153: length of this stabilization phase ?

We clarified the stabilization phase in the manuscript as follows:

*Only after their stabilization phase the LCS output is eligible for measurements of their respective target pollutant (Gäbel et al., 2022). The stabilization phase observed in the LCS outputs was shorter than one day. The first 24 hours of all LCS data were thus removed and not considered for this study.*

➢ Line 155: coma could be removed.

Done.

➢ Line 157: The 3 of O3 should be in subscript.

Done.

➢ Line 165: Are the daily means for LCS based on the hourly values or on the raw values ? The end of this paragraph suggest that the daily means has been calculated using hourly values. Did you check the impact on the data ?

We have clarified this in the manuscript as follows in lines 164-168:

*Raw LCS measurements and reference measurements given by the AEMS were aggregated to hourly means for LCS calibration. Calibrated PM2.5 measurements were aggregated to daily means. Daily means were required for the performance evaluation of the low-cost particulate matter sensor SAG-SPS30 based on the technical specification developed by CEN (CEN/TS 17660-2:2024, 2024) and the test protocol developed by EPA (Duvall et al., 2021a).*

➢ Line 183: This PM sensor sentence seems to me to be not in the right paragraph as the PM data has been discussed on the previous one.

We moved line 183 to the previous paragraph to line 168:

*Calibrated PM2.5 measurements were aggregated to daily means. Daily means were required for the performance evaluation of the low-cost particulate matter sensor SAG-*

*SPS30 based on the technical specification developed by CEN (CEN/TS 17660-2:2024, 2024) and the test protocol developed by EPA (Duvall et al., 2021a). The SAG-SPS30 provides outputs in mass concentrations by default.*

> ➢ Line184-189: This explanation could maybe be moved a after the first paragraph of 2.4 where the use of T and RH in the calibration models is explained. It was somehow confusing to me to read first that the data from the BME280 were not used to then see that they are finally used. Only on a second read I pay attention to the fact that the BME280 data were not used for the gas sensors.

The purpose of the paragraph is to emphasize that mass concentrations are required for sensor evaluation according to CEN/TS 17660-1:2021 and to specify which meteorological data we considered in order to calculate mass concentrations as accurately as possible. Therefore, we would prefer to keep these lines in the data treatment section, as the contents of the full paragraph are too closely interwoven.

To prevent confusion, we adjusted the lines 184-189:

*We exclusively used low-cost meteorological data from the Bosch BME280 sensors as input for the calibration models (Sect. 2.4). To calculate mass concentrations from the output of the calibration models we did not rely on BME280 meteorological data, but used the weather station data, because the former are highly biased due to solar radiation. The bias stems from solar heating of the AELCM units, which could not be mitigated by the integrated fan, as it causes an exchange of air between the inside and outside, failing to reduce the heating effect. It is planned to upgrade the AELCM units with radiation shields in the future to reduce the effect of solar radiation on the low-cost meteorological measurements.*

> ➢ Table 1: the first row is not the easiest to read, in particular for O3 and NO2 as there is not a clear separation between the T (end of O3) and VNO2 (beginning of NO2).

We improved the readability of the table:

**Table 1.** Model variables for the development of the calibration functions based on Multiple Linear Regression (MLR), Ridge Regression (RR), Random Forest (RF) and Extreme Gradient Boosting (XGB).

| Calibration Model | $O_3$ Model (Features / Target) | $NO_2$ Model (Features / Target) | CO Model (Features / Target) | $PM_{2.5}$ Model (Features / Target) |
|---|---|---|---|---|
| MLR | $V_{OX}$, $V_{NO2}$, $V_{CO}$, RH, T, $V_{OX} * T$ / $AEMS_{O3}$ | $V_{NO2}$, $V_{CO}$, RH, T, $V_{NO2} * T$ / $AEMS_{NO2}$ | $V_{CO}$, RH, T, $V_{CO} * T$, $\frac{(V_{CO})^2 - 1}{2}$ / $AEMS_{CO}$ | SPS30, RH, T, log(SPS30) / log($AEMS_{PM2.5}$) |
| RR | $V_{OX}$, $V_{NO2}$, $V_{CO}$, RH, T / $AEMS_{O3}$ | $V_{NO2}$, $V_{CO}$, RH, T / $AEMS_{NO2}$ | $V_{CO}$, RH, T / $AEMS_{CO}$ | SPS30, RH, T / $AEMS_{PM2.5}$ |
| RF | $V_{OX}$, $V_{NO2}$, $V_{CO}$, RH, T / $AEMS_{O3}$ | $V_{NO2}$, $V_{CO}$, RH, T / $AEMS_{NO2}$ | $V_{CO}$, RH, T / $AEMS_{CO}$ | SPS30, RH, T / $AEMS_{PM2.5}$ |
| XGB | $V_{OX}$, $V_{NO2}$, $V_{CO}$, RH, T / $AEMS_{O3}$ | $V_{NO2}$, $V_{CO}$, RH, T / $AEMS_{NO2}$ | $V_{CO}$, RH, T / $AEMS_{CO}$ | SPS30, RH, T / $AEMS_{PM2.5}$ |

➢ Line 218: what do you mean by merging the data by hour ? is it the mean calculation ?

We aligned the hourly reference station data with the hourly raw LCS data by matching timestamps. We think it is redundant to mention this, since time alignment is the standard procedure when comparing a reference method with a candidate method. Therefore, we removed "*and merging the data by hour*" in line 218.

➢ Line 395: you should mention in the previous paragraph 2.7 Performance metrics and target values that the measurement thus the evaluation has been carried out only for a urban background site whereas the CEN document ask for different testing site, for example a rural site for O3.

I've added this information in Section 2.7 at line 369:

*It should also be noted that the LCS evaluation was performed only at a single urban background site (AEMS), whereas the technical specifications by CEN call for evaluations at different sites, for instance, testing NO$_2$ sensors at traffic and background sites.*

▪ Figure 8, 9, 10 and 11: I would advice the authors to write the title of the different graphs on a clearer way, at a first look, it is not easy to see the difference between each plot.

We adjusted the title position and title size of each of the figures mentioned to enhance readability. The adjustments can be seen further down below:

[Figure]

**Figure 8.** Calculated REU values for MLR calibrated $O_3$ LCS hourly data belonging to the test periods (TP1–TP7, 10 June 2022–11 January 2023) of AELCM009 and AELCM010. The calibration variants are single training (ST) (top row, left: AELCM010, right: AELCM009) and extended training (ET) (bottom row). The extended training is characterized by ET variants of 1, 2 and 3 months for each AELCM box. Horizontal dashed lines describe the data quality objectives ($O_3$ Class 1 DQO = 30 %, Class 2 DQO = 75 % and Class 3 DQO = 200 %). The vertical dashed line describes the limit value for $O_3$ (LV = 120 µg m$^{-3}$). The fitted smooth curve (red) is based on a generalized additive model (GAM). Data density is shown through colour, where darker colours express lower data density and brighter colours express higher data density.

[Figure]

**Figure 9.** Calculated REU values for RF calibrated $O_3$ LCS hourly data belonging to the test periods (TP1–TP7, 10 June 2022–11 January 2023) of AELCM009 and AELCM010. The calibration variants are single training (ST) (top row, left: AELCM010, right: AELCM009) and extended training (ET) (bottom row). The extended training is characterized by ET variants of 1, 2 and 3 months for each AELCM box. Horizontal dashed lines describe the data quality objectives ($O_3$ Class 1 DQO = 30 %, Class 2 DQO = 75 % and Class 3 DQO = 200 %). The vertical dashed line describes the limit value for $O_3$ (LV = 120 µg m$^{-3}$). The fitted smooth curve (red) is based on a generalized additive model (GAM). Data density is shown through colour, where darker colours express lower data density and brighter colours express higher data density.

[Figure]

**Figure 10.** Calculated REU values for MLR calibrated PM$_{2.5}$ LCS daily data belonging to the test periods (TP1–TP7, 11 June 2022–6 January 2023) of AELCM009 and AELCM010. The calibration variants are single training (ST) (top row, left: AELCM010, right: AELCM009) and extended training (ET) (bottom row). The extended training is characterized by ET variants of 1, 2 and 3 months for each AELCM box. Horizontal dashed lines describe the data quality objectives (PM$_{2.5}$ Class 1 DQO = 50 %, Class 2 DQO = 100 % and Class 3 DQO = 200 %). The vertical dashed line describes the limit value for PM$_{2.5}$ (LV = 30 µg m$^{-3}$). The fitted smooth curve (red) is based on locally estimated scatterplot smoothing (LOESS). Data density is shown through colour, where darker colours express lower data density and brighter colours express higher data density.

[Figure]

**Figure 11.** Calculated REU values for RF calibrated $PM_{2.5}$ LCS daily data belonging to the test periods (TP1–TP7, 11 June 2022–6 January 2023) of AELCM009 and AELCM010. The calibration variants are single training (ST) (top row, left: AELCM010, right: AELCM009) and extended training (ET) (bottom row). The extended training is characterized by ET variants of 1, 2 and 3 months for each AELCM box. Horizontal dashed lines describe the data quality objectives ($PM_{2.5}$ Class 1 DQO = 50 %, Class 2 DQO = 100 % and Class 3 DQO = 200 %). The vertical dashed line describes the limit value for $PM_{2.5}$ (LV = 30 µg m$^{-3}$). The fitted smooth curve (red) is based on locally estimated scatterplot smoothing (LOESS). Data density is shown through colour, where darker colours express lower data density and brighter colours express higher data density.

---

## Author Comment (AC2)

**Response to comments from Referee #2**

**Citation:** https://doi.org/10.5194/egusphere-2025-2677-RC2

Thank you for your review and valuable comments. Our responses and revisions, which we believe will further enhance the quality of the paper, are presented below. The comments from Referee #2 are provided in black, our responses appear in brown, and the revised or newly added text in the manuscript is shown in *italics*.

This manuscript shows different options for calibration of LCS, in particular O3 and PM2.5. The goal is to show a tradeoff between the model accuracy based on an initial training with a dataset (in terms of duration) and recurrent recalibrations.

The discussion is interesting, and it is an open question. Notice that about this topic there are many issues to be considered for this problem, with regard to the initial dataset (in terms of quality, range, duration, sampling frequency, locations for deployments), models used for calibration (statistical ones or based on AI (machine learning, deep learning)), sensor types and features (gas, cross sensitivity, fabrication (Electrochemical, Metal OXide (MOX) sensor, NDIR and/or optical, aging effect) to name a few. Nevertheless, the authors focus on sensors O3 (Alphasense Ox-B431) and PM2.5 (Sensirion AG SPS30) and using 4 different models (MLR, RR, RF, XGB) for calibration.

Thank you for emphasizing the common issues and challenges that need to be considered in low-cost sensor (LCS) calibration, many of which we aim to address through recurrent calibration and, consequently, through continuous data quality assurance.

To clarify why we focused only on these two sensor technologies (electrochemical gas sensors and optical particle sensors): In our initial work (Gäbel et al., 2022), we tested LCSs based on different technologies to identify the most suitable ones for developing our own low-cost air pollution monitoring system. Based on the raw data quality and calibration results using the common multiple linear regression (MLR) method, we found that electrochemical sensors provided the most promising results for the measurement of ozone ($O_3$), while the Sensirion SPS30 (optical particle sensor) stood out in terms of performance compared to the other LCSs we investigated. Therefore, we decided to focus on these two sensor technologies. In the case of the SPS30, we did not explore other optical particle sensor candidates for the measurement of $PM_{2.5}$, as its performance was satisfactory, and we retained it for the latest, more advanced version of the Atmospheric Exposure Low-Cost Monitoring (AELCM) box.

In the present paper we investigated gas sensors from another manufacturer (Alphasense), which are based on electrochemical gas sensor technology, as a consequence of our findings (Gäbel et al., 2022) and other literature about Alphasense sensors. We applied additional calibration models, but our main focus was on recurrent calibration and its impact on performance. The study considers the recommendations of the U.S. EPA (United States Environmental Protection Agency) and European technical specifications (CEN/TSs) approved by CEN (European Committee for Standardization) for LCSs providing a novel perspective on sensor calibration design by using both as guidance to evaluate overall sensor performance and to investigate the suitability of the introduced LCS as supplemental tools for air quality monitoring.

Next, you have the suggested Comments (C) to improve your manuscript:

C1.- The title should be clearer and more specific including key words such as tradeoff, O3 and PM2.5

We would use tradeoff as one of the keywords for this study, but we would not include it directly in the title.

We suggest the following title change:

"Recalibration of low-cost $O_3$ and $PM_{2.5}$ sensors: Is it worth it?"

C2.-The study is carried out with 2 sensos O3 (Alphasense Ox-B431) and PM2.5 (Sensirion AG SPS30). The selection should be justified and motivated: why these ones? are these the more common, more reliable, price vs quality ratio, etc.? The authors should provide a survey (a study of state of art) about this. This information is very useful for the reader.

In addition, in Section 2.1, the name of the sensors for O3 and PM2.5 and their abbreviations (AS-B431, SAG-SPS30) as well as their features should be placed in a table to ease reading.

Thank you for the suggestions. We added more information and a new table based on the Reviewers input.

Line 122 – 125:

[revised manuscript text omitted]

C3.- The references are bit confusing. Not sure if it is the proper format and they are correctly compiled (not linked with reference section). For instance, (Gäbel et al., 2022), you cannot find it directly in the reference list. Although in a double lookup you can assume that it refers to a paper in Sensors MDPI from the same authors.

Also, an update of these references is welcome, with more recent ones.

Yes, we reference Gäbel et al. (2022), which is our earlier publication about the AELCM box in Sensors MDPI.

We adjusted the output style of the references to improve readability in the section "References" (Indentation and line spaces). References in the manuscript are easier to find now in the section "References". All references in the manuscript are included in this section.

We added some more recent literature, kept the relevant references and removed older references where it seemed appropriate.

C4.- Figure 1 is a bit confusing. Maybe a flow diagram of the proposal of the manuscript (the tradeoff between training duration and recalibration) should be better.

Done.

We would like to keep our original Figure 1 and present both figures side by side to make our methodological approach even clearer.

[Figure]

[Figure]

*Figure 1: Schematic representation of the pairwise calibration strategy and calibration model development as a flow diagram (top) and a time series scheme (bottom) using two LCS measurement systems (AELCM009 and AELCM010) showing the single training period (ST, 11 January–10 June 2022) and the extended training period (ET) as well as the numbered one-month test periods (TP) for each LCS measurement system. The thickness of the coloured lines in the flow diagram visually represents the amount of training data used for ET of the calibration model compared to ST.*

C5.-In my opinion, the analysis of 2 different deployments (AELCM009 and AELCM010) is interesting, to see the behavior (variability) between the different sensors.

But, the content of this manuscript could improved in a more comprehensive way. It could be carried out by using the whole dataset, and running on this dataset the different variables of the tradeoff: x= duration of initial training, y=recalibration time. Based on (x,y) you can plot the different metrics (R2, RMSE, REU,…) or a cost function (this is mentioned later in C11)) as a heatmap (in 3D plots), in stead of using a fixed training of 5 months, with extended periods of 1 months, and with recalibration with different periods. A heatmap should be easier to understand and see the optimum, rather than Figures 2-4 and 5-7. Notice that these figures are ambiguous and unclear. Also, the caption is bit redundant except 1, 2 or 3 months.

Besides, it should be noted that usually, the datasets have a higher sampling frequency, usually 10 min (or even lower), rather than 1 hour. It should be explained. Even, the sampling frequency could be a new variable to be considered in the tradeoff, instead of 1 hour as default.

Carotenuto et al. (2023) provide a literature survey about the topic of low-cost air quality monitoring networks for long-term field campaigns. They highlighted that in most cases, LCS networks are still only used for test applications or specific projects, most often not even lasting one year and that there is a lack of long-term efforts aiming at routinely monitoring air quality conditions.

To help encourage such long-term initiatives and stimulate interest among potential sensor end users such as local environmental agencies that also have permanent access to calibration equipment, we deliberately incorporated the recent test protocols from the U.S. EPA and CEN into our study. By applying the recommended performance metrics and performance targets from these protocols, our aim was to support practical decision making by stakeholders considering deeper involvement in air sensor projects, rather than to conduct an in-depth statistical analysis like suggested in C5 in the second paragraph.

We also wanted to avoid obscuring our key messages for end-use communities, centered on reaching performance targets and attaining the highest possible sensor tiers. This tier-based concept is easier for end-users and stakeholders to understand, especially for those who usually have the infrastructure and resources to maintain low-cost sensor networks over the long term and who ultimately need to be convinced of their value.

In our opinion, the approach we have chosen and the form of display (2D circular bar plots and REU plots) to check the achievement of performance targets and sensor tiers are very good from an end-user and practical perspective and also for the scientific community. We work with air sensor data ($O_3$, $PM_{2.5}$) and performance thresholds for RMSE, $R^2$, Intercept and slope and the relative expanded uncertainty (REU) at the limit value of $O_3$ and $PM_{2.5}$ as suggested by EPA test protocols and CEN test protocols, respectively.

We are specifically highlighting in our plots (Fig. 2-4 and Fig. 5-7), when a target is fulfilled (non-hatched bars in circular bar plots) and under which circumstances (Calibration model, single training (ST), extended training (ET) variant, AELCM box). Calibration model performances are ordered from highest to lowest in each test period (TP). Because of the manifold of aspects (Calibration model, ST, ET variant, AELCM box, time periods, error metrics and so on), which should be displayed in a single plot, and the question how recurrent

calibration should be designed, splitting figures by ET variants (1 month, 2 months, 3 months) is the most sensible choice in our opinion.

A further reason is, that an ET variant defines when an AELCM box needs to be exchanged with its partner AELCM box in situ. This is indicated through the curved lines in Fig. 2-4 and Fig. 5-7 (dashed: AELCM009, non-dashed: AELCM010). We also prefer 2D circular bar plots instead of 3D plots, because we can display TPs in a clocklike manner, which is an elegant way to communicate sensor performance over time in our opinion.

Our AELCM measurement systems have a sampling frequency of 4 seconds, as mentioned in line 119. We clarified it more in line 119:

*The upgrades also involved the increase of the sampling frequency for each AELCM sensor from 10 seconds to every 4 seconds.*

Hourly and daily means of LCS measurements were used to comply with the evaluation requirements of the CEN and EPA test protocols. We clarified that in line 164 till 168:

*Gas sensor measurements were aggregated to hourly means, while $PM_{2.5}$ sensor measurements were aggregated to daily means. This was required for the performance evaluation of LCSs according to the technical specification developed by CEN (CEN/TS 17660-1:2021, 2021; CEN/TS 17660-2:2024, 2024) and the test protocol developed by EPA (Duvall et al., 2021a, Duvall et al., 2021b). As a result, gas measurements and $PM_{2.5}$ measurements given by the AEMS were aggregated to hourly and daily means, respectively.*

[Figure]

[Figure]

(a)                                   (b)

*Figure 2. Photographs of the AEMS and AELCM units (AELCM009 and AELCM010), which are mounted on the fence next to the AEMS: (a) the stationary air and climate measurement station of the Chair for Regional Climate Change and Health, Faculty of Medicine, University of Augsburg; and (b) the housing and interior view of the engineered AELCM units.*

C7.- Section 2.4 requires a better description and detail of the models used. This can be summarized in a table with a short description and reference. Additional information could be interesting such as the library used, hyperparameters used (if needed), is there overfitting in the machine learning models? etc.

In Table 1, the target (in features/target) is not necessary if it is the same name of the model (on each column). Also, it should be recommended for clarity to show only the 2 models that you are using: O3 and PM2.5.

The tuned hyperparameters of our calibration models are provided in Table S3 of our Supplement. We added additional details and descriptions of the calibration models in Table S3 for interested readers and added the used R libraries. We refer to this table in the manuscript. Furthermore, we revised line 224 as follows:

*The selected and tuned model hyperparameters for RF, XGB and RR can be found in the supplement as well as more detailed information on the calibration models and used R packages (Table S3).*

Furthermore, we added additional information about the purpose of the mlr3 package, as we believe the relationship between mlr3 and the R packages listed in Table S3 may not be clear to readers. The mlr3 framework enables us to use models from multiple libraries through a single, unified interface for training, testing and evaluation. We revised line 223 to clarify the role of the mlr3 package:

*The mlr3 package and mlr3 ecosystem provide a framework for regression tasks and a unified interface for working with various learning algorithms, including the calibration models used in this work.*

The Reviewer raised concerns about overfitting; therefore, we added additional information in line 234 to clarify how we addressed overfitting during the calibration model building process:

*An out-of-sample (OOS) method following a repeated holdout strategy (Gäbel et al., 2022) was used to identify calibration models with good performance and optimally tuned hyperparameters, as estimated by their performance on the holdout data.*

We revised Table 1 as suggested by the Reviewer and moved the information about the $NO_2$ and CO models to the Supplement.

The reason the targets were initially all placed outside the column names is that we apply a specific transformation to a target of a single calibration model. Therefore, we wanted to be consistent in our display of information. This calibration model is the MLR-based calibration model for $PM_{2.5}$ sensor measurements (last column). We removed the other targets and added an asterisk to Table 1 explaining why this one target is retained in the table.

*Table 1. Model variables for the development of the calibration functions based on Multiple Linear Regression (MLR), Ridge Regression (RR), Random Forest (RF) and Extreme Gradient Boosting (XGB).*

| Calibration Model | $O_3$ Model Features | $PM_{2.5}$ Model Features [Target] |
|---|---|---|
| MLR | $V_{OX}$, $V_{NO2}$, $V_{CO}$, RH, T, $V_{OX}$ * T | SPS30, RH, T, log(SPS30) [log($AEMS_{PM2.5}$)]* |
| RR | $V_{OX}$, $V_{NO2}$, $V_{CO}$, RH, T | SPS30, RH, T |
| RF | $V_{OX}$, $V_{NO2}$, $V_{CO}$, RH, T | SPS30, RH, T |
| XGB | $V_{OX}$, $V_{NO2}$, $V_{CO}$, RH, T | SPS30, RH, T |

*\* This target is shown because it is transformed in the MLR calibration model configuration.*

Table S3. Description of the employed calibration models.

| Calibration Model | Description | Tuned Hyperparameters | R package | Reference |
|---|---|---|---|---|
| Extreme Gradient Boosting | • Decision tree-based ensemble machine learning method
• employs the gradient boosting framework
• Boosting is the concept of producing a strong learner from weak learners
• predictions are created from weak learners that continuously develop over the mistakes of the former learners | nrounds
eta
max_depth
lambda
alpha | xgboost | Mienye, I. D., & Sun, Y. (2022). A Survey of Ensemble Learning: Concepts, Algorithms, Applications, and Prospects. IEEE Access, 10, 99129–99149. https://doi.org/10.1109/access.2022.3207287

Zounemat-Kermani, M., Batelaan, O., Fadaee, M., & Hinkelmann, R. (2021). Ensemble machine learning paradigms in hydrology: A review. Journal of Hydrology, 598, 126266. https://doi.org/10.1016/j.jhydrol.2021.126266 |
| Random Forest | • tree-based ensemble machine learning method that uses decision trees as base-learners
• employs the bagging technique to build multiple decision trees using bootstrapped samples
• the bagging technique generates random samples with replacements from the input data and trains the decision trees from the samples
• predictions are created from the trained decision trees | mtry
sample.fraction
min.node.size
num.trees | ranger | Mienye, I. D., & Sun, Y. (2022). A Survey of Ensemble Learning: Concepts, Algorithms, Applications, and Prospects. IEEE Access, 10, 99129–99149. https://doi.org/10.1109/access.2022.3207287

Zounemat-Kermani, M., Batelaan, O., Fadaee, M., & Hinkelmann, R. (2021). Ensemble machine learning paradigms in hydrology: A review. Journal of Hydrology, 598, 126266. https://doi.org/10.1016/j.jhydrol.2021.126266 |
| Multiple Linear Regression | • regression method, which models linear relationships using least squares estimation
• linear combination of features (also called independent or explanatory variables), which are weighted by coefficients, to predict the target or dependent variable
• Assumptions:
  o linear relationship between features and target
  o residuals are normally distributed and independent
  o constant variance of residuals (Homoscedastic)
  o no outlier
  o no or a lack of multicollinearity | – | stats | Uyanık, G. K., & Güler, N. (2013). A Study on Multiple Linear Regression Analysis. Procedia - Social and Behavioral Sciences, 106, 234–240. https://doi.org/10.1016/j.sbspro.2013.12.027

Wilks, D. S. (2011). Statistical methods in the atmospheric sciences (Vol. 100). Academic press. |

| Calibration Model | Description | Tuned Hyperparameters | R package | Reference |
|---|---|---|---|---|
| Ridge Regression | • linear least squares regression method augmented by L2 regularization to address the bias-variance trade-off
• can be viewed as penalized regression
• Multiple linear regression is the simple non-regularized case of ridge regression | s | glmnet | Wanishsakpong, W., & Notodiputro, K. A. (2024). Comparing the performance of Ridge Regression and Lasso techniques for modelling daily maximum temperatures in Utaradit Province of Thailand. Modeling Earth Systems and Environment, 10(4), 5703–5716. https://doi.org/10.1007/s40808-024-02087-z

Nowack, P., Konstantinovskiy, L., Gardiner, H., & Cant, J. (2021). Machine learning calibration of low-cost NO2 and PM10 sensors: non-linear algorithms and their impact on site transferability. Atmospheric Measurement Techniques, 14(8), 5637–5655. https://doi.org/10.5194/amt-14-5637-2021

Asilevi, P. J., Dzidzorm, E. N., Boakye, P., & Quansah, E. (2025). Nitrogen dioxide (NO2) Meteorology and predictability for air quality management using TROPOMI. Npj Clean Air, 1(1). https://doi.org/10.1038/s44407-024-00003-4 |

C8.- Abbreviations are repeated many times. As a general rule for abbreviations, define them once and use them always, except in the abstract.

Besides, a glossary at the end of the paper should be interesting.

Done. We did adjustments to our manuscript to respect the general rule for abbreviations.

We added a list of abbreviations.

Appendix A: List of abbreviations

| | |
|---|---|
| AELCM | Atmospheric Exposure Low-Cost Monitoring |
| AEMS | Atmospheric Exposure Monitoring Station |
| $AEMS_{XX}$ | Concentration of a specific air substance measured by the AEMS |
| AQD | Air Quality Directive of the European Union |
| AS | Alphasense |
| AS-B431 | Alphasense B-Series electrochemical sensor for $O_3$ |
| AS-B43F | Alphasense B-Series electrochemical sensor for $NO_2$ |
| AS-B4 | Alphasense B-Series electrochemical sensor for CO |
| CEN | European Committee for Standardization |
| CET | Central European Time |
| CO | Carbon monoxide |
| DQO | Data quality objective |
| EC | Electrochemical |
| EPA | United States Environmental Protection Agency |
| ET | Extended training |
| GDE | Guide for the demonstration of equivalence |
| LCS | Low-cost (air) sensor |
| MLR | Multiple Linear Regression |
| MOS | Metal oxide semiconductor |
| $NO_x$ | Nitrogen oxides |
| NSIM | Non-regulatory supplemental and informational monitoring |
| $O_3$ | Ozone |

| OOS | Out-of-sample |
|---|---|
| $PM_{2.5}$ | Particulate matter (Particles that are 2.5 microns or less in diameter) |
| $PM_{10}$ | Particulate matter (Particles that are 10 microns or less in diameter) |
| $R^2$ | Coefficient of determination |
| REU | Relative expanded uncertainty |
| RF | Random Forest |
| $RH_{XX}$ | Relative humidity of a specific BME280 sensor in an AELCM unit |
| RMSE | Root-mean-squared error |
| RR | Ridge Regression |
| Rs | Spearman rank correlation |
| $SO_2$ | Sulfur dioxide |
| SAG | Sensirion AG |
| SAG-SPS30 | Sensirion AG optical particle sensor for $PM_1$ and $PM_{2.5}$ |
| $SPS30_{XX}$ | Particulate matter concentration of a specific SAG-SPS30 in an AELCM unit |
| ST | Single training |
| $T_{XX}$ | Temperature of a specific BME280 sensor in an AELCM unit |
| TP | Test period |
| TS | Technical specification |
| UTC | Coordinated Universal Time |
| $V_{XX}$ | Net voltage of a specific AS sensor in an AELCM unit |
| WHO | World Health Organization |
| XGB | Extreme Gradient Boosting |

C9.- In addition to Table 2 (with the stats of the dataset for 1 day), why do not you plot the stats for the whole period (1 year?) and/or plot their value over the time?

Is it correct 36º in Augsburg?

Also, you can also include in Table 2 the same stats for all the features (variables) of your dataset (AEMSxx, Vxx).

These statistics are not for a single day but cover a specific timespan. For example, in the second column of the first row, you will see 11/01/22 – 11/01/23. Due to unfortunate formatting and the lack of space, this wasn't immediately clear, but all calculated statistics for the variables in column 1 are based on an entire year of data. We adjusted the table description of Table 2 and added the following to clarify:

*Statistics based on the hourly means of the atmospheric variables measured by the AEMS from January 2022 to January 2023.*

Plotted values over time related to Table 2 can be found in the Supplement of this work (Figure S1-S4).

According to Germany's National Meteorological Service, the Deutscher Wetterdienst (DWD), the DWD station in Augsburg recorded a daily maximum temperature of 35.9 °C on 20/07/2022, which is close to the daily maximum temperature of 35.65 °C that we measured on the same day. Therefore, the daily maximum temperature given in Table 2 appears to be correct. We obtained the station data from the DWD Climate Data Center, which provides open data: https://www.dwd.de/EN/climate_environment/cdc/cdc_node_en.html

Thank you for the suggestion to include the statistics for the raw output data in the table. We initially considered this but decided not to include it in the manuscript. In our view, presenting raw sensor signals, such as the sensors' net voltages, would not add meaningful value and would obscure the main message of Table 2. The purpose of Table 2 is to characterize the environmental conditions during the collocation period and to provide a first impression of the information content of the raw sensor signals. In our opinion, this is already achieved through the Spearman rank correlation (Rs), which illustrates the relationship between the station measurements and the raw sensor signals.

C10.- Conclusions are too long. You could simplify them add more relevant conclusions, since it is well known that with these LCS, recalibration is always required.

Besides, both in the abstract and in conclusion, you should highlight your contribution.

We shortened and simplified the section "Conclusions", focusing on the relevant conclusions. We also highlighted our own contributions in the abstract and conclusion.

Our Abstract changes to highlight our own contributions to the community:

Line 9 – 11:

*In this study, we demonstrate how widely used air sensors (OX-B431 and SPS30) for the relevant air pollutants ozone ($O_3$) and fine particulate matter ($PM_{2.5}$) by two manufacturers (Alphasense and Sensirion) should be recalibrated for real-world monitoring applications.*

Line 12 – 14:

*We use multiple novel test protocols for air sensors provided by the United States Environmental Protection Agency (EPA) and the European Committee for Standardization (CEN) for evaluative guidance and to identify possible applications for OX-B431 and SPS30 sensors.*

Line 21 – 24:

*We investigated different recalibration cycles using a pairwise calibration strategy, which is an uncommon method for recurrent LCS calibration. Our results indicate that a regular in-season recalibration is required to obtain the highest quantitative validity and broadest range of applications for the analyzed LCSs, with monthly recalibrations appearing to be the most suitable approach.*

Line 27 – 29:

*Compared to one-time pre-deployment sensor calibration, in-season recalibration can broaden the scope of application for a LCS (indicative and non-regulatory supplemental measurements)*

*and must be considered by the end-use communities, if certain real-word applications are supposed to be performed reliably by LCSs and to achieve sufficient information content.*

Our updated and adjusted conclusions (Line 724 – 800):

*In an attempt to consistently provide air sensor performance by a pair of $O_3$ and $PM_{2.5}$ LCSs (AS-B431 und SAG-SPS30) suitable for supplementing official air quality monitoring networks, an still uncommon approach for recurrent sensor calibration was explored during a yearlong collocation campaign at an urban background station next to the University Hospital Augsburg, Germany.*

[revised manuscript text omitted]

C11.- As mentioned before in C5, if you plot heatmap find other suggestions to visualize the results:
1. **Error-vs-time curves**: plot RMSE(t) for different recalibration strategies. This shows how quickly accuracy decays and how recalibration recovers it.
2. **Heatmap**: x-axis = initial training duration ($T_0$), y-axis = recalibration interval (days). z = a metrics (RMSE, R2, …). This visually shows regions where short initial training + frequent recalibration ≈ long initial training + infrequent recalibration.
3. **Pareto frontier / cost-accuracy plot**: x-axis = operational/calibration cost, y-axis = long-term mean RMSE. Mark strategies on the plot.
4. **Bar chart**: number of recalibrations vs mean RMSE for each $T_0$.
5. **Time-to-failure distributions**: for threshold-triggered policies, plot histogram of detection delays.
6. **Uncertainty band plots** (error ± CI) to show statistical significance between strategies.

Thank you for your detailed suggestions.

We would prefer to keep our circular bar plots for the visualization of our results. The reasoning for that is explained in our response to C5.